# Consistent negative response of US crops to high temperatures in observations and crop models

Bernhard Schauberger[1], Sotirios Archontoulis[2], Almut Arneth[3], Juraj Balkovic[4,5], Philippe Ciais[6], Delphine Deryng[7], Joshua Elliott[7], Christian Folberth[4,8], Nikolay Khabarov[4], Christoph Müller[1], Thomas A.M. Pugh[3,9], Susanne Rolinski[1], Sibyll Schaphoff[1], Erwin Schmid[10], Xuhui Wang[11,12], Wolfram Schlenker[13] & Katja Frieler[1]

High temperatures are detrimental to crop yields and could lead to global warming-driven reductions in agricultural productivity. To assess future threats, the majority of studies used process-based crop models, but their ability to represent effects of high temperature has been questioned. Here we show that an ensemble of nine crop models reproduces the observed average temperature responses of US maize, soybean and wheat yields. Each day > 30 °C diminishes maize and soybean yields by up to 6% under rainfed conditions. Declines observed in irrigated areas, or simulated assuming full irrigation, are weak. This supports the hypothesis that water stress induced by high temperatures causes the decline. For wheat a negative response to high temperature is neither observed nor simulated under historical conditions, since critical temperatures are rarely exceeded during the growing season. In the future, yields are modelled to decline for all three crops at temperatures > 30 °C. Elevated $CO_2$ can only weakly reduce these yield losses, in contrast to irrigation.

[1] Climate Impacts and Vulnerabilities, Potsdam Institute for Climate Impact Research (PIK), 14473 Potsdam, Germany. [2] Department of Agronomy, Iowa State University, Ames, Iowa 50011, USA. [3] Institute of Meteorology and Climate Research-Atmospheric Environmental Research (IMK-IFU), Karlsruhe Institute of Technology, 82467 Garmisch-Partenkirchen, Germany. [4] International Institute for Applied Systems Analysis, Ecosystem Services and Management Program, Schlossplatz 1, A-2361 Laxenburg, Austria. [5] Department of Soil Science, Faculty of Natural Sciences, Comenius University in Bratislava, 84215 Bratislava, Slovak Republic. [6] Laboratoire des Sciences du Climat et de l'Environnement, Institut Pierre-Simon Laplace (IPSL), 91191 Gif sur Yvette, France. [7] University of Chicago and ANL Computation Institute, Chicago, Illinois 60637, USA. [8] Department of Geography, Ludwig Maximilian University, 80333 Munich, Germany. [9] School of Geography, Earth & Environmental Science and Birmingham Institute of Forest Research, University of Birmingham, Birmingham B15 2TT, UK. [10] University of Natural Resources and Life Sciences, Vienna, Feistmantelstrasse 4, 1180 Vienna, Austria. [11] Laboratoire de Météorologie Dynamique, Institute Pierre-Simon Laplace, 95005 Paris, France. [12] Sino-French Institute of Earth System Sciences, College of Urban and Environmental Sciences, Peking University, Beijing 100871, China. [13] School of International and Public Affairs, Columbia University, New York, New York 10027, USA. Correspondence and requests for materials should be addressed to B.S. (email: schauber@pik-potsdam.de).

Crops grow best within specific intermediate temperature intervals. Excessive frost or heat are detrimental to physiological processes and, eventually, yield levels. Under climate change episodes of high temperature are expected to increase in frequency and duration. This could threaten regional productivity in already susceptible areas[1–4]. There are a number of statistical approaches that allow for separating effects of high temperatures on observed yields from other sources of variability that are not correlated with them over time. Rainfed maize, soybean and cotton yields in the US have been shown in statistical studies to decline non-linearly with temperatures above ~30 °C (ref. 5). Wheat in the US responds negatively to frost in fall or heat in spring; the reduction due to high temperature is lowered by increased rainfall[6]. Maize yields in Africa decline strongly with temperatures >30 °C, in particular under lack of water[7]. Senescence of irrigated wheat in India is accelerated by temperatures >34 °C (ref. 8). But these statistical models are agnostic about the underlying mechanisms, which are important to understand to help farmers better adapt to high temperatures. Process-based crop models, in contrast, provide an implementation of physiological crop growth processes. They model complex responses of crop yields to climate change, accounting for weather fluctuations on (sub-)daily time scales. In particular, they allow for varying responses in terms of the phenological state of the crop, for interactions between the atmospheric $CO_2$ concentration (henceforth [$CO_2$]), temperature, precipitation and other weather variables, and delayed effects of precipitation due to soil water storage.

High temperatures, which are defined as temperatures >30 °C within this study, affect crop yields by direct and indirect effects. High temperatures can cause water stress through depletion of soil water and increased atmospheric water demand[9–12], which leads to a closing of stomata to avoid desiccation (thereby reducing the uptake of $CO_2$) and also to an enhanced root growth at the expense of above-ground biomass. High temperatures can also directly damage enzymes and tissues[13–15], impair flowering[10,16], trigger oxidative stress[17], lead to precocious maturity and senescence (resulting in less time for accumulating biomass[18,19]) or lower net photosynthesis rates due to lower carbon (C) assimilation and/or higher respiration rates[20–22]. By using one site-based crop model for three corn-growing locations in the US corn belt it has been shown that the observed high-temperature effects on maize yield are largely mediated by changes in water supply and demand rather than by direct damage to the plant tissues[9]. The critical role of water availability to cope with high-temperature stress is also shown for African maize, where negative effects on yields >30 °C double under drought conditions[7].

Here we apply the statistical approach by Schlenker and Roberts[5] to simulated yields from process-based models to test their representation of observed negative high-temperature effects on a spatially aggregated level. We analyse maize, soybean and wheat, which are US staple crops occupying 62% of the 2010 harvested area in the US[23] and 33% globally[24]. To test the sensitivity to water availability, we make separate comparisons for predominantly rainfed or irrigated counties. In addition, we derive the average response to high temperature under future

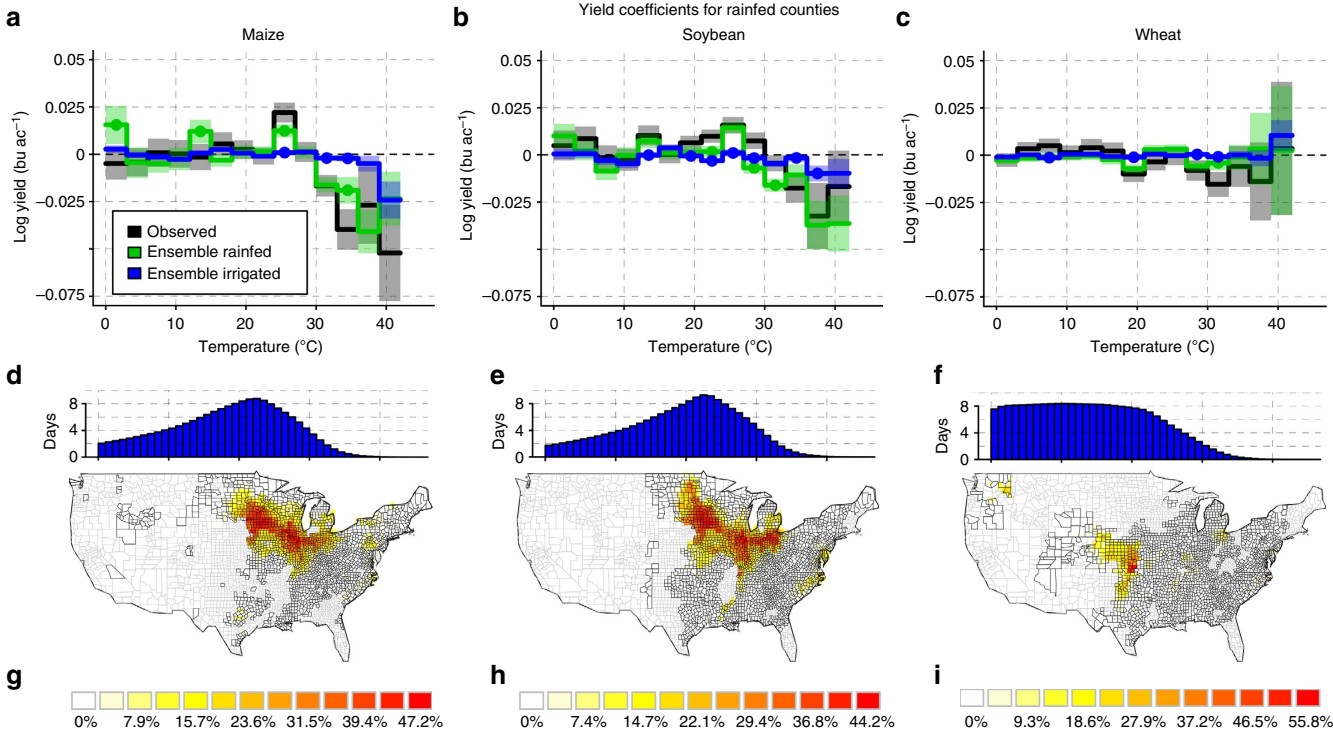

**Figure 1 | Comparison of statistically estimated effects of temperatures on observed and simulated US yields in rainfed counties.** Columns are maize (**a**,**d**,**g**), soybean (**b**,**e**,**h**) and wheat (**c**,**f**,**i**). **a**–**c** show regression coefficients and **d**–**f** show the histogram of times spent in individual temperature bins as the sum of times derived for each grid point across the growing seasons. **g**–**i** show rainfed counties (black outlines) with their per cent land-use share (colours) of the respective crop (for wheat only counties with predominantly winter wheat). Black lines in **a**–**c**: coefficients $\gamma_h$ derived from log-transformed observed yields (Methods; equation (1)). Green/blue lines: coefficients of the ensemble median rainfed/irrigated simulated yields. Estimates are derived by a panel regression of US county data, where the considered crop is grown under predominantly (>90%) rainfed conditions. Shaded areas represent 95% confidence intervals. Simulated coefficients are marked by coloured dots if they are significantly different from the observed coefficients (confidence intervals do not overlap).

(2071–2099) climate conditions and higher levels of atmospheric $CO_2$ under Representative Concentration Pathway RCP8.5. While the empirical approach in ref. 5 does not account for the effects of higher $[CO_2]$ on future yields, it is explicitly represented in process-based models. We find that the crop models of our ensemble include the most relevant mechanisms of high-temperature-induced yield loss under current climate, in particular a water-dependent temperature response in agreement with observations. Elevated $CO_2$ cannot be confirmed as a safeguard of yields under high temperatures, in contrast to previous assumptions. A shift of temperatures from beneficial to detrimental in a narrow temperature range can already induce large crop losses—which can reliably be assessed by current models.

## Results

**Models capture observed yield responses to high temperatures.** The considered ensemble of nine Global Gridded Crop Models (GGCMs; eight for wheat) is able to closely reproduce the observed average response of rainfed crop yields ($\gamma_h$, Methods, equation (1)) to time spent in different temperatures from 0 to 42 °C (Fig. 1, green and black lines). The statistical model estimates the changes in yield if the crop is exposed to temperatures within individual intervals for one day. A value of $\gamma = -0.04$ as, for example, derived from the observed maize yields for the temperature interval from 33 to 36 °C means that one additional day at these temperatures would reduce the yield by $1 - \exp(-0.04) \approx 4\%$. The results are robust against the form of the statistical analysis (step function or piecewise linear, Supplementary Figs 1–3; principal component regression, Supplementary Fig. 4; Supplementary Note 1), fertilizer input (Supplementary Figs 5–7) and growing season assumptions (Supplementary Figs 8–11). In the main text, we therefore only

show results for crop model-specific default representations of present-day management conditions[25] and fixed growing seasons following Schlenker and Roberts[5] (Methods).

Only 7 out of 42 coefficients significantly diverge between the regression models for observed and simulated yields (95% confidence intervals do not overlap). The confidence intervals become larger at higher temperatures, owing to less time exposed to these temperature bins. Responses for the individual models can be found in Supplementary Fig. 12; see also Supplementary Note 2. The temperature threshold of roughly 30 °C (maize and soybean peak at the 24–27 °C interval, which is one temperature bin lower than earlier estimates for maize[5]) is in close agreement with values deduced from field experiments[7,9,26,27]. In contrast to maize and soybean, wheat shows no clear temperature response pattern or decline with high temperature (Fig. 1c), neither for observed nor for simulated yields. Not all models are able to simulate winter wheat, so we excluded those which only simulate spring wheat (Methods). Given the close agreement between observed and simulated yield average responses, we use the process-based models to identify the mechanism behind the decline in yields.

**Models suggest water stress as major cause of yield declines.** The coefficients derived from the median of the simulated ensemble under the assumption of full irrigation (blue lines in Fig. 1) significantly diverge from the coefficients derived from simulations assuming rainfed conditions (green lines) at 7, 8 and 4 out of 14 temperature bins each for maize, soybean and wheat, respectively (cf. also the modified scaling and correlation of coefficients in Supplementary Figs 13—15; Supplementary Note 3). Full irrigation reduces the negative effect of temperatures $>30$ °C. Although a detrimental effect of very high temperatures $>39$ °C seems to occur even for irrigated maize, the

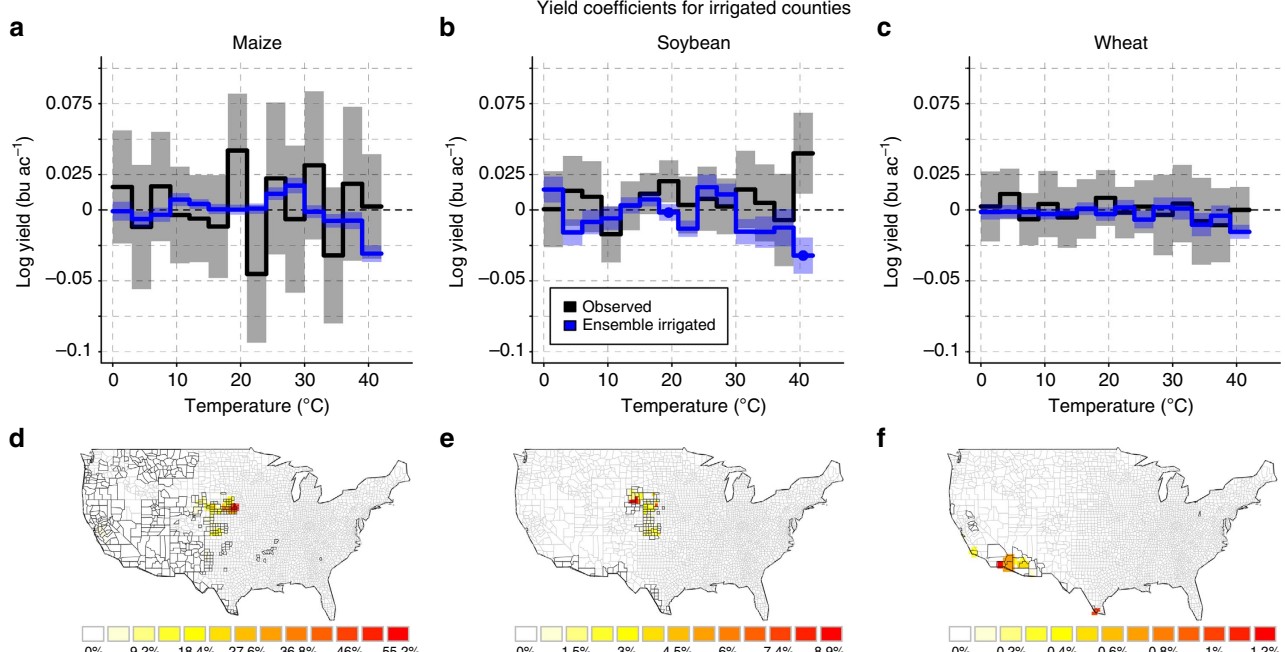

**Figure 2 | Comparison of statistically estimated effects of temperatures on observed and simulated US yields in irrigated counties.** Unconstrained irrigation is assumed on the irrigated areas specified by MIRCA2000 (ref. 24). Columns are maize (**a,d**), soybean (**b,e**) and wheat (**c,f**). **a–c** show regression coefficients and **d–f** show irrigated counties (black outlines) with their per cent land-use share (colours) of the respective crop. Counties are considered as irrigated if $>75\%$ of the crop-specific-harvested area is irrigated. Black and blue lines in **a–c** represent coefficients $\gamma_h$ for observed and simulated yields, respectively. Shaded areas are 95% confidence intervals. Results for individual models are shown in Supplementary Fig. 33.

interpretation of this single coefficient may be misleading due to the small number of data points. In irrigated counties (Supplementary Fig. 16) neither the observations nor the simulations show a strong decline in yield coefficients at high-temperature intervals (Fig. 2; Supplementary Note 4). The confidence intervals for irrigated counties are larger, partly due to fewer observations (Methods), making the statistical model estimates noisy. The crop model ensembles for maize and soybean show a yield decline with temperatures $>33\,°C$ and $30\,°C$, respectively, but less pronounced than in the rainfed case. All confidence intervals in the high-temperature range are close to 0 except for $39–42\,°C$.

The crop model simulations assuming full irrigation on rainfed areas show a significantly higher evapotranspiration (ET; Supplementary Fig. 17) and a significantly higher biomass accumulation (Supplementary Fig. 18; Supplementary Table 3) than the rainfed runs. All models simulate shorter growing seasons with higher average temperatures for maize and soybean. For wheat the effect can be confounded by vernalization, which is delayed under higher temperatures, such that only a majority of the models shows a decrease. The average decline in length for each additional degree of average growing season temperature over the period 1980–2010 is $\sim7.4$ days for rainfed maize, 5.6 days for soybean and 1.3 days for wheat, respectively. This decline is equal or higher under irrigated conditions in the same counties (equal for maize, but 9% and 46% higher for soybean and wheat, respectively).

**Models suggest that CO₂ only limitedly attenuates yield loss.** The interaction of temperature, water and $[CO_2]$ plays an important role for future yields under global warming[17]. To assess this we apply the panel regression to simulated future yields in rainfed counties under climate change (RCP 8.5). We use an ensemble of six GGCMs (five for wheat), whose models overlap with the historical ensemble above (Methods). Four settings are analysed: rainfed conditions and fixed present-day $[CO_2]$ levels, rainfed conditions and elevated $[CO_2]$ (803 p.p.m. as 2071–2099 mean), full irrigation and fixed $[CO_2]$, and full irrigation and elevated $[CO_2]$. Rainfed yields continue to exhibit a pronounced decline at high temperatures, even under elevated $[CO_2]$ (Fig. 3, solid and dashed green lines).

Under climate change and the associated shift of growing season temperatures into the critical range $>30\,°C$ wheat also shows a decline in yields under rainfed conditions (Fig. 3c). The signal can strongly be reduced with irrigation (blue lines) for all crops. The bottom part of each panel in Fig. 3 shows the shifts of temperature distributions over the fixed growing season into warmer ranges for the future (red solid line) when compared to the historical period (1980–2010, grey dashed line). We do not consider irrigated counties for this analysis since the historical response shows large uncertainties.

The median rainfed yields of the future model ensemble show a generally reduced temperature sensitivity caused by elevated $[CO_2]$, also at higher temperatures for maize and wheat, evidenced by the smaller absolute coefficient values over the whole temperature range. This holds for the individual models, too (Supplementary Figs 19–21). But these reductions are not significant for any of the crops over the whole temperature range (confidence intervals overlap everywhere). In contrast, the coefficients for irrigated yields are nearly equal for fixed and elevated $[CO_2]$ at all temperatures, for all three crops. They diverge significantly from the rainfed coefficients at 9 out of 42 coefficients, in particular in the temperature range $>30\,°C$.

Elevated $[CO_2]$ significantly reduces actual ET and increases biomass and yield under rainfed and irrigated conditions for all three crops (Supplementary Figs 22–25; Supplementary Table 4). For maize, however, the biomass increase with elevated $[CO_2]$ is only marginal under irrigated conditions (4.6%) in comparison with soybean (35.2%) and wheat (19.4%). For soybean the reduction in ET at elevated $[CO_2]$ is only marginal (1.4%) under rainfed conditions.

## Discussion

We applied a statistical model to detect the temperature response of observed and simulated county yields in the US. We showed that the considered ensemble of nine process-based crop models is capable of reproducing the observed detrimental effects of high temperatures on rainfed maize and soybean crops. For wheat neither observations nor simulations show a decline in the historical period. The close agreement between rainfed simulations and observations and a strongly reduced yield decline with ample water supply in the models allows us to

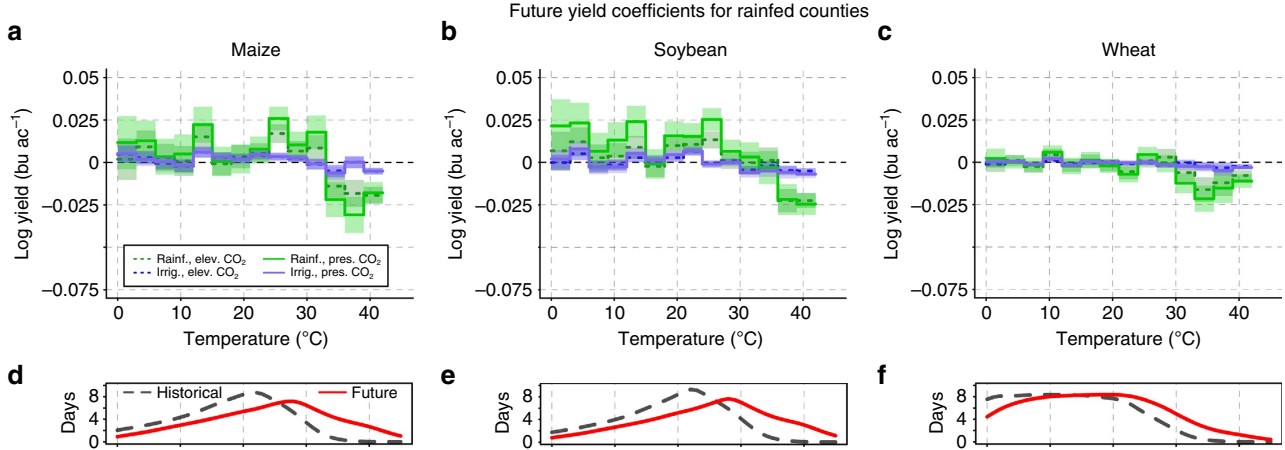

**Figure 3 | Simulated yield responses to temperature under future climate change in rainfed counties.** Columns are maize (**a,d**), soybean (**b,e**) and wheat (**c,f**). **a–c** show regression coefficients and **d–f** display temperature histograms for the historic (dashed grey) and future (solid red) periods; future climate is evaluated over 2071–2099 based on RCP8.5. Green tone lines in **a–c** are ensemble yield responses to temperature under rainfed conditions. Blue tone lines are ensemble yield responses under irrigation. Solid lines are derived with fixed present-day $[CO_2]$, while dotted lines include elevated $[CO_2]$ according to RCP8.5. Shaded areas are 95% confidence intervals. Rainfed counties are defined in Fig. 1.

conclude that irrigation lowers the temperature sensitivity of all three crops. In the future, the models suggest a negative response of maize, soybean and wheat to high temperatures even under elevated [$CO_2$]. A future shift of temperatures from beneficial to detrimental may reduce crop yields substantially even without considering the effect of extremely high temperatures.

Negative effects of high temperature on wheat would be expected at temperatures >30 °C (ref. 26). Under historical conditions wheat was usually harvested before high-temperature stress occurred, or the stress occurred during non-sensitive phenological stages. The occurrence of temperatures >30 °C per growing season is, on average, higher for maize (10.8 days) and soybean (13.1 days) compared with wheat (6.0 days). Field trial data in Kansas[6] has shown sensitivity of wheat to temperatures above 34 °C in spring, which we do not observe for the larger geographic coverage and given the rare occurrence of such spring heat events in the past.

The close agreement of high-temperature responses of observed and simulated yields allows for an investigation of the underlying mechanism of the yield decline. In particular, the threshold response >30 °C, which is not natively implemented in the models, is a prerequisite for this investigation. The dampening effect of irrigation on the temperature response of yield supports the hypothesis that temperature-induced water stress is the main driver of the observed yield decline at temperatures >30 °C, in line with the study by Lobell et al.[9] Atmospheric water demand increases with temperature as an immediate effect. In addition, water supply from soil to plant gradually decreases due to depletion of soil water stocks from sustained high ET. Both factors can lead to water stress for crops, where the stomata gradually close to prevent water loss and therefore preclude the diffusion of $CO_2$ into the cells. This leads to a reduced gross photosynthesis rate. All GGCMs considered here represent both the immediate (stomatal closure) and progressive (soil water depletion) effects of temperature (model characteristics in Supplementary Table 1). In addition, crops respond to water stress by enhanced root growth at the expense of above-ground biomass and yield; this effect is included in eight of the nine models (Supplementary Table 1). The critical role of water supply at high temperature is further supported by the yield response curves for observed yields from predominantly irrigated counties, where no clear temperature response is visible. Yet this yield response in irrigated counties is rather noisy due to few observations (Methods). But our conclusions mainly rely on the (counterfactual) irrigated yield response in rainfed counties, where a larger panel allows for robust assessments. Troy et al.[28] have recently shown that irrigation attenuates the yield impacts of several climate-extreme indices, which is in accordance with our findings. Thus reduced gross photosynthesis rate, triggered by reduced $CO_2$ inflow under water stress, constitutes a major pathway for yield decline under high-temperature conditions without sufficient water supply (first point from the effects listed in the introduction).

Yet the existence of temperature-induced water stress does not necessarily preclude other negative effects of high temperatures (other points from the list above). The first three of the alternative explanations (direct damage to enzymes and tissue, impaired flowering and oxidative stress) are not represented in the considered crop model ensemble (except impaired flowering in one model, PEGASUS). That the ensemble is nevertheless able to reproduce the observed decline in yields at temperature levels of 30–36 °C suggests that these three effects are not the main causes of the observed decline in yields in this temperature range at this spatial coverage. Direct damage to enzymes, tissues or reproductive organs is only expected at higher-temperature levels (35–37 °C for maize and 35–39 °C for soybean; refs 26,27) than

the thresholds identified here. The actual leaf temperature could deviate from the surrounding air temperature, since water scarcity precludes a transpirational cooling of the leaves. Yet, none of the considered models explicitly accounts for leaf temperature differences to ambient air. Furthermore, there is evidence that irrigation does not only reduce the perceived temperature for the plant, but also the actual temperature over large regions[29–31]. This effect is not considered in the crop models. But given the agreement between observations and simulations, a direct damage seems to be of minor relevance for the general shape of the temperature response at the range considered here. Increasing oxidative stress can arise from higher levels of photorespiration or higher uptake rates of ozone ($O_3$), whose concentrations tend to increase with temperature[32]. A potential increase in photorespiration is expected to be less pronounced in $C_4$ plants like maize[13,17,22], which is not supported by the observational data showing a particularly pronounced decline in maize yields. For $O_3$, irrigation could even increase its damaging effects, since more available water allows the stomata to open wider, which would let more $O_3$ in ref. 33. Thus, the first three alternative pathways do likely not explain the observed yield reduction under rainfed conditions and its alleviation under irrigation.

In contrast, the crop models do simulate shorter growing seasons with increasing temperature(Supplementary Table 5). The phenological development of crops is mainly controlled by temperature, such that (non-adapted) crop plants would have less time for gaining biomass and yield if the growing season shortens. This could explain yield declines with high temperature. But in the model ensemble the growing season lengths shorten equally or even more for irrigated yields than rainfed yields. So a shorter maturity time does not explain why there is no reduction in yields for irrigated conditions. In addition, observations show that maturity may even be delayed, instead of advanced, by high temperatures[9,34].

Seven of nine models include a direct effect of temperature on maintenance respiration (Supplementary Table 1), and the other two have a lower radiation use efficiency under high-temperature stress. Net biomass gain is the difference between gross photosynthesis and plant respiration, such that an increased respiration can lead to lower biomass and yield. Respiration data are not available from the model ensemble considered, but the relative share of respiration to assimilation is expected to increase with high temperature[22] and water stress[15]. An evaluation of the 2003 European heat wave, however, found a decreasing respiration under heat and drought conditions[21]. Respiration equations in the models are influenced by temperature only, not by water supply. Therefore increased respiration under high-temperature stress does not explain why there is no yield decline under irrigation, in particular since models have no cooling effect of transpiration on perceived temperature. Together with the ambiguous response of respiration to high temperature or drought stress, we suggest that increased respiration is not a primary reason for the yield decline under high temperatures within the range analysed here.

The statistical approach is sensitive to yield losses induced by extremely high temperatures, despite their low relative abundance in the data set (Supplementary Fig. 26; Supplementary Note 5). At the same time, the direct damage to enzymes, tissues or reproductive organs expected in these temperature ranges is not represented in the crop models (see above). Thus, the agreement between observations and simulations indicates that damage directly induced by extremely high temperatures is of minor relevance in the historical sample on the large spatial scale of our study. Damages in the observed yields could be limited if temperatures occurred in noncritical periods of the growing

season. But in the considered sample extreme temperatures mainly occurred in the middle and last phase of the growing season, in which anthesis and grain filling mostly occur (Supplementary Fig. 27). Both these processes are known to be critically sensitive to high temperatures[8,10,20,22,35–37]. In addition, a sensitivity test regarding the timing of the exposure and the definition of the growing season has not revealed a significant difference in the associated responses to extreme temperatures[5]. Evaporative cooling may have reduced leaf temperatures to lower values than air temperatures, which are used as predictor in the regression model. The latter aspect is not represented in the crop models and requires further work to quantify the role of evaporative cooling, as a protection mechanism[38,39]. In addition, harvests may have been adjusted to avoid exposure to extremely high temperatures, an effect not represented in the exposure times used in our analysis. Yet, given the abundant total number of such extremely high temperatures in our data set (41,580 days >36 °C for maize, 70,934 for soybean and 34,200 for wheat), we argue that the latter explanation is less relevant. The agreement between the observed and simulated temperature sensitivities found for the historical sample does not imply that models capture all processes relevant under future climate change, where direct temperature-induced damages may become more relevant. However, based on the regression coefficients derived from the historical observations and temperature shifts projected for the end of the century by HadGEM2-ES under RCP8.5, increasing exposure to temperatures in the range from 30 °C to 36 °C alone implies yield losses of 49% for maize, 40% for soybean and 22% for wheat (Table 1). Our analysis suggests that crop models reliably simulate temperature effects in this range. A further test of the reliability of future projections of yield losses could be achieved by assessing regions that are already warmer today, or of field experiments where temperatures are artificially increased[40,41].

Assuming that the crop models are able to capture the relevant mechanisms that lower yields at high temperatures, as discussed above, we continue to investigate the simulated future interactions between high temperature, water supply and $CO_2$ concentrations. We only consider rainfed counties (maps in Fig. 1), since the estimates of the statistical model in irrigated counties (Fig. 2) are too noisy to base any extrapolation on them. An elevated concentration of $CO_2$ is reported as a yield-increasing factor for most plants[12,32]. It tends to increase crop water-use efficiency (gain of carbon per unit of water lost) and maintain higher levels of soil moisture. Observations have confirmed that $CO_2$ fertilization is usually more efficient under drought conditions, even for $C_4$ plants such as maize[17,42]. But the only insignificant differences in high-temperature response of yields with elevated [$CO_2$] suggest

that elevated [$CO_2$] has a limited potential to buffer against detrimental effects of temperature-induced water stress on crop yields. These findings do not contradict beneficial effects of $CO_2$ on yield, in particular when integrating over the growing season (Supplementary Fig. 25). But they suggest that episodic temperature-induced water stress cannot be attenuated effectively with higher [$CO_2$] alone. In particular for soybean elevated [$CO_2$] leads to more biomass (larger leaf area), which in turn increases transpiration needs (Supplementary Fig. 23). Thus, the amount of water required by soybean under elevated [$CO_2$] is similar to that under fixed [$CO_2$], despite higher water-use efficiency. As a consequence the plant responds in a similar way to the water stress triggered by elevated temperature. Thus, a strong biomass increase under elevated [$CO_2$] prevents an ameliorating effect of [$CO_2$] under episodic temperature-induced water stress (similar conclusions are derived in refs 9,17,43,44). For wheat ($C_3$) and maize ($C_4$) the biomass increase under elevated [$CO_2$] is smaller (Supplementary Figs 22 and 24). Therefore, the temperature-induced water stress can better be attenuated with higher [$CO_2$] in these two crops when compared with soybean, but still not significantly. These hypotheses are based on model results in rainfed counties only, where a robust response to temperature is visible for simulated rainfed and irrigated yields (Fig. 1), and could guide further experiments on the role of $CO_2$ under high-temperature stress.

Estimated yield responses under high levels of global warming should not be interpreted as predictions, since the GGCM simulations do not commonly account for potential adaptation options. The implementation of management and thus adaptation options differs between models. For example, fertilizer application rates were held constant (PEGASUS, pDSSAT and pAPSIM) or adjusted flexibly according to nitrogen stress (EPIC-IIASA, EPIC-BOKU and GEPIC). The choice of cultivars was only allowed to change trough time in PEGASUS, LPJ-GUESS and limitedly in GEPIC. Thus, the ensemble response to temperature exposure represents the average response across a range of different management assumptions. Individual farmer's options to adapt to more frequent temperature stress could dampen negative yield responses—though the extent may be limited[5,45].

The effects of $CO_2$ on yield formation are taken from the individual models' best estimate, which have partly been calibrated against experiments to capture yield responses to $CO_2$ (ref. 46). There is a discussion that crop models may overestimate yield response to elevated levels of $CO_2$ (refs 42,47). Furthermore, an adequate sensitivity of the models to temperature or water supply does not imply any conclusions on the adequacy of the $CO_2$ effect in models. Caution needs to be exercised also when extrapolating historical temperature

**Table 1 | Contribution to yield changes by different temperature ranges.**

| Crop | Time | Yield change factors | | | | Future yield loss below 36 °C |
|------|------|------|------|------|------|------|
| | | **Below 30 °C** | **30–36 °C** | **Above 36 °C** | **Total** | |
| Maize | Historical | 1.80 | 0.73 | 0.96 | 1.27 | 49% |
| | Future | 1.62 | 0.41 | 0.47 | 0.31 | |
| Soybean | Historical | 2.84 | 0.88 | 0.95 | 2.37 | 40% |
| | Future | 2.12 | 0.71 | 0.59 | 0.89 | |
| Wheat | Historical | 0.93 | 0.91 | 0.99 | 0.84 | 22% |
| | Future | 0.85 | 0.78 | 0.94 | 0.62 | |

Numbers are yield change factors for different temperature ranges that modify the base yield resulting from intercept, precipitation, county-fixed effects and time trends. The total column indicates the product of all temperature exposures >0 °C on yield. The last column indicates yield loss expected from a shift of temperature exposures only within the 0–36 °C range (calculated with equation 2).

responses into the future, as temperature effects that are of minor relevance in the past may become more important in the future, in particular in temperature ranges not observed in the historical data set. Direct crop damages from extremely high temperatures (for example, 40 °C) are usually not represented in current crop models and would have to be improved before assessing crop responses to these extremes in the future[48]. But already the shift towards higher temperatures from beneficial to detrimental (histograms in Fig. 3), without considering extreme temperatures, poses a strong challenge for rainfed crop production (Table 1). An increase of irrigated areas or irrigation efficiency to overcome (parts of) the negative consequences would be effective. Yet potential constraints of water availability have to be accounted for refs 49–51.

Some of the models in our historical and future ensembles belong to model families with a shared history of development. Specifically, the three EPIC-based models (EPIC-Boku, EPIC-IIASA and GEPIC) share an identical model core, but have distinct assumptions on input and crop-specific parameters, and the two LPJ-type models (LPJ-GUESS and LPJmL) share the same photosynthesis approach, but diverge, for example, in allocation or crop-specific assumptions. Yet a shared model history does not prescribe a similar response to environmental conditions. This is exemplified by the different responses of models even of the same families (Supplementary Figs 8–10), which is comparable to differences between models of distinct families. As a consequence we assume the confidence intervals and model ensembles to be unbiased with respect to model families.

Our study provides insight into high-temperature-induced mechanisms of yield losses at an aggregate scale and thus constitutes a complement to field-based or experimental studies. The latter allow for a direct control of temperature and confounding variables, but are necessarily restricted to few locations and have until now only sparse coverage of the whole US[40,41,52]. Therefore experimental bottom-up and top-down regression approaches are both necessary to elucidate crop responses under climate change. The applied statistical approach allows extracting average yield responses to exposure to different temperature bins across a large spatial area with varying small-scale management conditions. As such it is particularly suitable for the evaluation of GGCMs rather designed to reproduce yields responses on large scale than to resolve fine-scale variations in management. It adds to well-established knowledge of yield responses to temperature that is derived from field and chamber experiments. The application of GGCMs may help us to explore adaptation options on large scales.

The crop models used here do not represent all potentially detrimental effects of high temperature. Short-term changes in management, such as fertilizer input, or diseases and pests also influence observed yield fluctuations[53], but are often not well documented and also not always represented in the models. But the simulations show a water-dependent temperature response that is in agreement with the observations. Therefore, we infer that the crop models include the most relevant mechanisms under current climate. Though extreme temperatures will become more important under climate change, and crop models will have to capture the associated effects[48], already the shift in the exposure times to temperatures in the range from 30 to 36 °C can induce large crop losses—which can reliably be assessed by current models. Despite the clear ensemble response, there are several cases where the combined temperature water effects are either under- or overestimated, and this behaviour should be investigated further in the process-based models. The accurate simulation of yield response to

temperature does not necessarily imply an accurate reproduction of observed yield time series, since other factors like management could mask them. We suggest further field experiments to assess our model-based hypothesis of a limited effect of elevated [CO$_2$] under water stress induced by high temperatures. In addition, models with an explicit representation of leaf temperature could help to deepen our understanding of the processes involved in yield decline under high temperatures and further improve crop projections under climate change.

## Methods

**Climate data.** Historical: we employed daily temperature (maximum and minimum) and precipitation data for the statistical model, and further weather variables for the yield simulations by the GGCMs, from the AgMERRA climate data set[54], covering the years 1980–2010. The weather data were spatially aggregated to 0.5° for the crop simulations[25]. We used the identical data set for the statistical analysis. Its spatial resolution is one order of magnitude coarser than in the original empirical study[5], which could result in less temperature extremes due to aggregation effects. But the slight deviation between the temperature distributions of the two data sets (Supplementary Fig. 29,30; Supplementary Note 6) only has a minor effect on the estimated coefficients (Supplementary Fig. 31). In addition, predicted yields from the regression model based on the AgMERRA data are in close agreement with the observed yields in terms of mean growing season temperatures (Supplementary Fig. 32). Future: all future model results (statistical and process-based) are forced by bias-corrected[55] climate projections from the HadGEM2 climate model under the RCP8.5 scenario at 0.5° spatial resolution. We applied only one climate model, instead of an ensemble, since we study relative temperature responses rather than absolute yield levels.

**Yield data.** Historical observed US county yields from 1980 to 2010 (to 2008 for wheat) were downloaded from the USDA Quick Stats tool[23]. Historical yield simulations were calculated under the default and harmnoN harmonization scenarios (differing in fertilizer input, growing season definition and irrigation choices, cf. ref. 25) by nine different crop models: EPIC-Boku, EPIC-IIASA (both, ref. 56), GEPIC[57], LPJ-GUESS[58], LPJmL[59], ORCHIDEE-crop[60], pAPSIM[61], pDSSAT[62] and PEGASUS[63]. All GGCMs are forced by the same climate input[54], which is also used to calculate the time of the growing season that is spent within the different temperature bins. Historical model yields were generated within the GGCM Intercomparison project[25] of the Agricultural Modelling Intercomparison and Improvement Project (AgMIP)[64]. Future yield simulations (years 2071–2099) were taken from the Inter-Sectoral Impacts Model Intercomparison Project (ISI-MIP)[65] Fast-Track data archive, once with CO$_2$ fixed at present-day levels (364–380 p.p.m. for all except pDSSAT which uses 330 p.p.m.) and once with elevated CO$_2$ (803 p.p.m. as 2071–2099 average). Yields from six models were available: EPIC-Boku, GEPIC, LPJ-GUESS, LPJmL, pDSSAT and PEGASUS. Note that model results for historical and future simulations were submitted at different times (future: 2011, historical: 2014 onwards); therefore, a direct comparison between the two responses is possibly biased due to differences in model versions. PEGASUS is excluded from both wheat ensembles, since it only simulates spring wheat. The crop models have not been calibrated against the observed temperature response used for validation here.

**Derivation of times spent in different temperature bins.** In analogy to ref. 5, we calculated the days spent in each 1° temperature bin during a fixed growing season (from March 01 to August 31 for both maize and soybean, and October 15 to July 15 for wheat) for each grid cell, using a sinus interpolation between daily minimum and maximum temperature. We then aggregated this data to county level with the MIRCA2000 land-use pattern[24], weighting by irrigated and rainfed shares, and considered only aggregated 3-K temperature bins as in ref. 5. In addition to the fixed growing season, the calculation was repeated for the model-specific growing seasons. For the future period from 2071 to 2099 the times spent in individual temperature bins were derived analogously, based on the bias-corrected climate projections.

**Regression model.** We pool the US county yields for each crop and irrigation setting to achieve a higher frequency of the rare high-temperature events in our data set (also pursued in ref. 28). A panel regression, implemented in R and following the procedure in ref. 5, was fitted separately to observed and simulated crop yields for all US counties, individually for rainfed and irrigated counties. A county was classified as rainfed or irrigated if its crop-specific area share was at least 90% rainfed or 75% irrigated, respectively. Mixed counties (rainfed share between 25 and 90%) were excluded. The following equation was applied for fitting:

$$\log Y_{it} = \alpha_0 + \sum_{h=0,3,6,\ldots}^{39} \gamma_h [\theta_{it}(h+3) - \theta_{it}(h)] + \mathbf{z}_{it}\delta + c_i + \varepsilon_{it} \qquad (1)$$

where $Y$ is yield, log the natural logarithm, $i$ the county and $t$ the year. $\theta_{it}(h)$ is the cumulative distribution function of days during the growing season spent at temperature h, and the $\gamma_h$ represent the estimated scaling coefficients shown in Figs 1–3. In addition, the model adjusts for a common intercept to all counties $\alpha_0$ and county-specific fixed effects $c_i$. Variations in precipitation (linear and quadratic) and state-specific time trends (linear and quadratic) to capture technological change are subsumed in $\mathbf{z}_{it}$ with the fitted scaling factors $\delta$. The residual error is described by $\varepsilon_{it}$; these error terms are allowed to correlate spatially as in ref. 5, estimated with the non-parametric method proposed by ref. 66, and applying a cutoff of $3°$ spatial distance. All temperatures $>39\,°C$ were subsumed into the same bin for $39$–$42\,°C$ (mean value before pooling is $40\,°C$ for all three crops), while the effect of temperatures $<0\,°C$ is captured by the fitted intercept. The total number of rows in the panels for historical observed rainfed maize, soybean and wheat are 42,648, 41,920 and 38,845, respectively, and 2,277, 719, and 149 county-year entries for irrigated counties. The total number of parameters to be fitted is $\sim 80$ for rainfed counties and $\sim 25$ for irrigated counties (depending on the number of states in the panel).

**Contribution of temperature shifts to yield losses.** We split the temperature distribution into three parts: $<30\,°C$ (no stress), $30$–$36\,°C$ (medium high temperature) and $>36\,°C$ (extreme high temperature; consistent with previous thresholds[8,35–37,67,68]). We calculate the relative contribution to yield for each of these parts by multiplying the coefficients estimated from observed yields with the historical or future exposure time for each $3\,°C$ bin. This results in change factors that modify the base yield resulting from intercept, precipitation, county-fixed effects and time trends. Yield loss by exposure shifts up to $36\,°C$ is then calculated with the ratio of these factors (equation 2).

$$\text{loss} = 1 - \frac{e^{\sum_{h=0,3,6,\ldots}^{33} \gamma_h \left[\theta_{avg}^{fut}(h+3) - \theta_{avg}^{fut}(h)\right]}}{e^{\sum_{h=0,3,6,\ldots}^{33} \gamma_h \left[\theta_{avg}^{hist}(h+3) - \theta_{avg}^{hist}(h)\right]}} \quad (2)$$

**Code availability.** All codes (R scripts) necessary to reproduce our results are available from the corresponding author on request.

**Data availability.** All data supporting the findings of this study are either public data sets, are available within the article and its Supplementary information files or are available from the corresponding author upon request.

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

## Acknowledgements

We thank Frank Wechsung from PIK Potsdam for insightful discussions. B.S. acknowledges support from the German National Academic Foundation. The work was supported within the framework of the Leibniz Competition (SAW-2013-PIK-5), from EU FP7 project HELIX (grant no. FP7-603864-2), EU FP7 Project LUC4C (grant no. 603542) and by the German Federal Ministry of Education and Research (BMBF, grant no. 01LS1201A1).

## Author contributions

B.S. and K.F. designed and performed the analysis and wrote the paper. W.S. originally designed the regression approach of yields to temperature exposure. All other authors contributed model results, helped to analyse data and commented on the manuscript.

## Additional information

**Competing financial interests:** The authors declare no competing financial interests.

