## [Peer Review File · Nature Communications]

Reviewers' comments:

Reviewer #1 (Remarks to the Author):

Review of "Consistent Negative Responses ... US crops ...crop models' by B. Schaubberger et al.

I find this manuscript unsuitable for publication in Nature.

Overall, the authors appear to confuse the term 'heat' with 'temperature', and mostly speak about mean temperature anomalies, while they claim their conclusions to be relevant to analyses of extreme heat events.

The list of heat-related impacts they provide on pg. 3, from a) -f), is in fact a mixed of direct physiological effects of extreme heat (oxidative stress; damage to enzymes; impaired flowering) and indirect ones from mean warming (precocious maturity and senescence; lower net photosynthesis; water stress).

Analyzing impacts of extreme events would require looking into the first set of dynamics listed above. Unfortunately, as Tab. 1 of the SI well shows, the crop models employed by the authors are unable to capture precisely such dynamics. Rather, the crop models used capture effects on yields of mean temperature anomalies.

So the authors are not really investigating impacts of extreme heat events at all.

Secondly, for comparing their simulation results, they choose statistical time series of observed yields in the US, which have been teased out to determine the effects of mean temperature anomalies in the US on overall crop development, yield, and the relations with water stress, including in cases with and without irrigation. Importantly, these time series, by the nature of the large scales they cover, contain very little to no information on the impact of extreme heat events on US yields.

Finally, given the two points above, it is no wonder that the authors find consistency between the simulations of heat impacts made with models that have no capacity to capture extreme heat events, with the negative (mean) temperature dependencies highlighted by a statistical analysis of historical yields. The consistency is precisely due to the fact that neither series, i.e., modeled or observed, contains relevant information on extreme heat events. All they point to is well-known dynamics of mean temperature anomalies and reduced yields that have nothing to do with the supposed subject matter of the paper.

Just to reinforce the point: one would expect exactly the opposite interaction between irrigation management and the impacts of extreme heat on physiology: such effects would actually be larger under irrigated conditions, since the physiological damage associated to them is absolute (damage to enzymes, flowering, etc.). Rainfed crop conditions, with associated complex water stress dynamics depending on the vagaries of precipitation, would instead confound the additional negative effects of extreme heat and produce effects that would be harder to notice. The fact that the authors find the opposite is another proof that what they really are considering are the impacts of mean temperature anomalies on crops via water dynamics -nothing to do with extreme heat events but that can nonetheless be well captured by the models employed-and reflect observations of mean crop-climate-water dynamics very well.

Thus there is no new information on extreme heat stress provided by this study, in fact mere a wrong analysis of facts leading to wrong conclusions. Furthermore, I strive to understand what use would a farmer have of the trivial facts discovered through this unnecessarily complex modeling machinery.

I would also suggest to avoid linking any effect on crop yield to food security considerations. Physiology and food security have very little in common and no logical link to each other (food security is determined by socio-economic and policy drivers well above and beyond crop function).

Reviewer #2 (Remarks to the Author):

The paper is an important contribution on an important topic. It's a paper that someone has needed to write for a long time -- so it's great that the authors have done so! -- and at least a subset of the results should be very useful to the broader debate about potential climate impacts on agriculture. I have one main concern, followed by some more minor stuff.

1. There seems to be an important disconnect between what appears to be the main goal of the paper - to show that crop models and observations (statistical models) give comparable results - and a bunch of the other results they show which (1) either look only at crop model results, or (2) show results that are fairly inconclusive with regard to consistency between models and observations.

For instance, the results section titled "Irrigation strongly attenuates the yield declines at heat (sic) temperatures" is somewhat confusing for at least 3 reasons: (1) the observed model for irrigated regions is very noisy (Fig 2a,b), meaning you can't really draw any conclusion here, (2) the crop models do suggest a clear reduction in yield above about 33C for corn and 30C for soy, and (3) the crop models do not appear to agree with each other (Fig S9b) about irrigated impacts at high temperatures.

If we have to be cautious about the historical irrigated area results, then we also have to be cautious about the projections using irrigated area. Again, if the article is about comparing statistical models to crop models, then you can't conclude that "Irrigation can overcome detrimental heat effects also in the future" (next section header), because the irrigated results for the statistical model are presumably noisy enough to admit really negative estimates as well as really positive ones. So the proper summary of the result here would seem to be: "Some crop models suggest that irrigation can overcome about half of the negative impacts for rainfed areas, but estimates from statistical models are too noisy to draw this conclusion".

Similarly on CO₂: we do not have an observational/statistical estimate of the fertilization effect (clearly very hard to get one, although people are trying). So again, are we just now supposed to believe the crop models on CO₂ because they got the rainfed effect "right", or should the authors again be clear that the CO₂ effect is from the crop models alone and thus should not be interpreted as something we know from observations. If it's the former, then the authors need to make this assumption very explicit - although it's hard to see how consistency on rainfed response under fixed CO₂ would convince us that the models get the CO₂ effect right.

More minor:

2. Is correspondence between statistical models and crop models a surprise, or have the latter been calibrated to the former? A little more detail on that would be useful.

3. Fig S6: this figure seems important but it's not clear what's plotted here. these are the coefficients from all the different temperature bins?

4. Table S3: could get high R² out of Fig S6 but sign could be wrong (e.g. see model plotted in blue). Maybe be clear about this? What are we to conclude from Fig S8?

5. Fig S9. caption appears to be wrong? panel b appears to show simulated rainfed (ensemble average is dotted line?) vs simulated irrigated, but caption says "rainfed observed versus irrigated simulated". or am I misunderstanding? the rainfed observed in panel a does not look like rainfed observed in panel b.

Reviewer #3 (Remarks to the Author):

This paper describes an analysis of observed data and a process-based crop model ensemble to (1) determine if such crop models can capture observed heat effects, and (2) to simulate future crop yields with projected future climates. The authors demonstrate that the crop model ensemble can reproduce observed heat responses, and that these models project declined yields in the future.

The paper is clearly written, and the references are appropriate.

Previous work has established that high temperatures are detrimental to crop yields, and that climate change is likely to bring higher temperatures. Previous studies have also shown that crop models simulated with future climate scenarios exhibit reduced yields. However, this paper is of interest to the crop modelling community due to (1) their analysis of a curated ensemble of process-based climate models, (2) their investigation of the salient heat and CO₂ processes captured (or not) by this ensemble, and (3) their model-based argument that temperature-induced water stress is the key process in heat stress.

There are two issues that should be dealt with before publication.

First, it should be made clear in the main paper that, being restricted by available data, the ensemble is comprised of a small number of distinct models, particularly when projecting future wheat yields. Page 1 of the SI states that these 9 models "have distinct histories of development and are therefore individual with respect to implementation of temperature and CO₂ responses." The GEPIC, EPIC-Boku, and EPIC-IIASA models are all of the same model family, as are LPJ-GUESS and LPJml. Since ORCHIDEE-crop and pAPSIM were excluded in the future yield projections, there were 4 distinct model families used for future maize and soybean simulations, and 3 for future wheat simulations. The reader needs to be aware of this when interpreting the medians and 95% confidence intervals in the main figures, especially given the differences between individual models at high temperatures (e.g., figure 2 vs figure S14), and given that the maize model with the strongest negative irrigated response at high temperatures (pAPSIM, fig. S18) was excluded from future projections.

Second, more work needs to be done to justify that the 0.5 degree spatial resolution can be used with their statistical method. Aggregating weather data can reduce local extremes, which are important with respect to crop heat stress. Perhaps this is why the interquartile ranges in figure S2 are all smaller for predicted than observed at high temperatures, for all crops? As the authors note, the AgMERRA data is an order of magnitude coarser than the data used by Schlenker and Roberts (S&R), and they are right to test the sensitivity of their coefficients to the climate data resolution. However, they do this by checking if the confidence intervals of their coefficients overlap with those of S&R. If confidence intervals do not overlap, there is statistical significance, but the opposite is not true.

-- Minor points -

Perhaps justify why an ensemble of crop models is required for the future yield analysis, but only a single climate model is necessary (HadGEM2). If HadGEM2 is best for US, why not weight crop models according to how well they match observations?

Replace 'most's with counts or percents (e.g., page 8: "Most models include...", page S5 "Individual models mostly ...", page 25 ".. and mostly even stronger so...").

It would be useful to the reader to include an explanation of why ORCHIDEE-crop, EPIC-Boku and pDSSAT can deviate from observed so much (figure S3). Same goes for figure S14.

'EPIC' is used as a model description throughout the SI (e.g., p. 21). Is it related to one of the 9 models in table S1?

Responses to the reviewer comments on “Consistent negative response of US crops to high temperatures in observations and crop models”

We want to thank the reviewers for their comments on our manuscript. We apologize for not having been clear enough on some terms and methods and have put emphasis on more clarity in the revised version of our manuscript. Individual responses to the comments raised by the reviewers (*written in Italic*) are given below.

Reviewer #1 (Remarks to the Author):

Review of "Consistent Negative Responses ... US crops ...crop models' by B. Schauburger et al.

Response: We want to thank the reviewer for his/her critical but important comments. We apologize for unclear terminology that may have led to misunderstanding of our results. In the following we discuss the individual points raised by the reviewer and state where we have adapted the manuscript for better understanding. We assume that there have been two principal misapprehensions: we consider direct *and indirect* crop damages from high temperatures, and we apply a regression approach that analyses temperature effects *on an aggregate scale with daily temperature resolution*.

I find this manuscript unsuitable for publication in Nature.

Overall, the authors appear to confuse the term 'heat' with 'temperature', and mostly speak about mean temperature anomalies, while they claim their conclusions to be relevant to analyses of extreme heat events.

Response: We are sorry for any confusion or misunderstanding arising from the usage of the terms “temperature” and “heat” in our manuscript. To avoid this, we have replaced “heat” by “high temperature” throughout the text. There are two types of different usages of the terms “heat” and “temperature” that could lead to misunderstanding of our analysis. The first is *physical*, where heat is the total energy of molecular motion or the transfer of thermal energy between bodies, while temperature is the average energy of molecular motion. The second is *agronomic*, where the reviewer might think of heat only as very high temperatures above e.g. 40°C where physiological damage to plants occurs. Yet in the existing agronomic literature “heat” is regularly used for temperatures above a threshold around 30-33°C^{1,2,3}.

The reviewer might also consider our analysis as unapt for capturing effects of hot days on crop yield. We would like to draw the attention to the statistical model we apply:

$$\log Y_{it} = \alpha_0 + \sum_{h=0,3,6,\dots}^{39} \gamma_h [\theta_{it}(h+3) - \theta_{it}(h)] + \mathbf{z}_{it}\boldsymbol{\delta} + c_i + \varepsilon_{it}$$

i.e., we estimate the coefficients γ_h describing the effect that times spent within the temperature interval $[h, h + 3]$ have on log-transformed county-level crop yields Y_{it} . We incorporate the entire distribution of temperatures between the daily minimum and maximum and count the hours spent in each interval, i.e. our results are not driven by daily average temperature. As these times vary from growing season to growing season and from county to county it becomes possible to estimate these individual effects and not just the effects of “mean temperature anomalies”. This is fundamental for our study and particularly allows for a separation of effects of exposure to high temperatures (above 30°C) from effects of exposure to lower temperatures. Other variables are explained in the main paper.

Furthermore we assume that the concepts of high temperature damage on crops seem to differ between the reviewer and us: while the reviewer seems to acknowledge only direct physiological damage like e.g. protein denaturalization we consider a broader range of possible impacts from high temperatures as listed in the introduction of the main paper. See more details on this in the next point.

The list of heat-related impacts they provide on pg. 3, from a) -f), is in fact a mixed of direct physiological effects of extreme heat (oxidative stress; damage to enzymes; impaired flowering) and indirect ones from mean warming (precocious maturity and senescence; lower net photosynthesis; water stress).

Response:

We agree with the reviewer that the different effects of temperature on crop yields could be grouped into two categories – one category describing directly destructive effects of ambient (or leaf) temperature such as the damage to enzymes, and another category where the effects are not directly destructive but accumulate over the growing season (leading e.g. to precocious maturity and senescence). The empirical model accounts for both effects by assuming a multiplicative relationship between the times spent in different temperature bins and overall yields. The mean temperature (anomaly) in a given growing season (which is how we interpret the reviewer’s concern regarding “mean warming”) is not a physiological variable. Plants only perceive and respond to immediate ambient temperatures, not to “mean temperatures“. All physiological responses are the result of direct environmental impacts on plant growth that possibly accumulate over the season. The statistical model provides a simplified representation of these instantaneous effects and their accumulation over the growing season. In particular, a shortening of the growing season due to earlier maturity and senescence can also be due to direct effects of high temperatures^{4, 5, 6}. Water stress is relevant on two time scales: an immediate effect through higher transpiration demands and a possible long-term effect through depleted soil water that cannot be transpired later^{1, 7}. The other four effects are immediate responses of the plants (see references in the main paper introduction). In conclusion, all six effects listed in the introduction can be caused by high temperature stress and are of agronomic importance.

Analyzing impacts of extreme events would require looking into the first set of dynamics listed above. Unfortunately, as Tab. 1 of the SI well shows, the crop models employed by the authors are unable to capture precisely such dynamics. Rather, the crop models used capture effects on yields of mean temperature anomalies.

Response:

None of the nine process-based crop models applied in our study uses „mean temperature anomalies” as input. They resolve the effects of ambient temperatures on different physiological processes at daily time steps (see SI Table S1). We agree with the reviewer that they do not resolve all of the six listed processes but clearly represent the temperature effects on growing season length (as a function of crop development, which proceeds according to the times spent at different temperatures), water scarcity, respiration, photosynthesis, leaf expansion and senescence, grain growth rate as well as temperature effects on soil water and nitrogen (in some models) processes which affect yields.

The statistical model applied to observed and simulated yield panels shows a close quantitative agreement between the observed and simulated temperature response. Therefore we conclude that the processes currently not explicitly represented in the models are of minor relevance to reproduce the observed temperature response, as we state in the discussion section.

So the authors are not really investigating impacts of extreme heat events at all.

Response:

We are sorry for the confusion that may be due to a different understanding of high temperature impacts on crops and the aggregate scope of the analysis. We hope that our responses and the language modifications in the whole manuscript clarify that we investigate direct and indirect impacts of high temperatures.

Secondly, for comparing their simulation results, they choose statistical time series of observed yields in the US, which have been teased out to determine the effects of mean temperature anomalies in the US on overall crop development, yield, and the relations with water stress, including in cases with and without irrigation. Importantly, these time series, by the nature of the large scales they cover, contain very little to no information on the impact of extreme heat events on US yields.

Response:

The objective of our paper is not to study the effects of individual high temperature events on physiological plant processes (like net primary productivity or evapotranspiration). Instead, we focus on the aggregated yield response that can be explained by the time crops were exposed to high temperatures⁸ and test to what extent this large-scale response can be reproduced by different crop models that resolve different processes relevant to the overall yield response.

The time series of observed yields and weather information is on a county level, i.e. much higher resolved than just national time series. Each annual county yield value represents the cumulative effect of growing conditions, in particular the temperature evolution, over the whole growing season. It represents the aggregated effect of ambient temperatures and other variables like e.g. precipitation. The statistical model applied here assumes that there is a multiplicative relationship between the average time spent within a specific temperature bin and annual yields. The temperature exposure times are calculated at each grid cell and weighted according to the fraction of the land area of the grid cell covered by the considered crop. We use the distribution of daily temperatures between the minimum and maximum to count what portion of the day a crop is exposed to a certain temperature range. Certainly the individual effects of these exposure times cannot be disaggregated from individual annual yield data. Therefore we sum the sub-daily exposure to various temperature ranges over the growing season. Since weather is random year-to-year, this panel variation allows us to identify the effect of various temperature ranges on crop yields from year to year and from county to county. Therefore these panel data contain information on the impacts of high temperatures on crop yields.

There are 12.9, 14.9 and 6.8 days above 30°C in the 1980-2010 average US growing seasons for maize, soybean and wheat, respectively (see histograms in Figure 1 in the main paper, and Figure S2 in the supplement). The considered default growing season length is 183 days for maize and soybean and 273 days for wheat, thus a fraction of 7.0, 8.1 and 2.5% of the growing season is a “high temperature” period. As wheat has a lower high-temperature-stress threshold than the other crops⁹ the definition of “high temperature” could also be lowered for wheat, such that the fraction of high temperature days increases.

Finally, given the two points above, it is no wonder that the authors find consistency between the simulations of heat impacts made with models that have no capacity to capture extreme heat events, with the negative (mean) temperature dependencies highlighted by a statistical analysis of historical yields. The consistency is precisely due to the fact that neither series, i.e., modeled or observed, contains relevant information on extreme heat events. All they point to is well-known dynamics of mean temperature anomalies and reduced yields that have nothing to do with the supposed subject matter of the paper.

Response:

We do not follow the interpretation of the reviewer, which we assume to come from a misinterpretation of our study as explained above. More specifically, we do incorporate daily temperature extremes (minimum and maximum), not just averages, and we study effects of high temperatures on yield at an aggregated level.

The close agreement between the temperature sensitivities of observed and simulated yields has not been expected, in particular since previous research has pointed to a limited capacity of crop models to simulate high temperature effects¹⁰. The response curve of yields to daily temperature exposure (as shown in Figure 1 of the main paper) is different from a simple correlation between yields and mean growing season temperature (as provided in SI Figure S3) that can suffer from confounding factors.

Just to reinforce the point: one would expect exactly the opposite interaction between irrigation management and the impacts of extreme heat on physiology: such effects would actually be larger under irrigated conditions, since they physiological damage associated to them is absolute (damage to enzymes, flowering, etc.). Rainfed crop conditions, with associated complex water stress dynamics depending on the vagaries of precipitation, would instead confound the additional negative effects of extreme heat and produce effects that would be harder to notice. The fact that the authors find the opposite is another proof that what they really are considering are the impacts of mean temperature anomalies on crops via water dynamics -nothing to do with extreme heat events but that can nonetheless be well captured by the models employed-and reflect observations of mean crop-climate-water dynamics very well.

Response:

We agree with the reviewer that when only considering direct physiological effects of high temperature the crop response may be more clearly visible under irrigation, when variations due to water stress are excluded. Yet, as stated above and in the manuscript, this is not the focus of our study. There is also a manifold of articles showing that irrigation (or ample water supply) buffers crops from high temperature stress^{11, 12, 13, 14}. When excluding water stress and looking only at direct physiological damages one would have to consider leaf temperatures, which are lower for irrigated yields (more water allows more transpirational cooling), possibly leading to less damage¹⁴ (and not larger ones, as the reviewer states). None of the global models in our ensemble considers leaf temperatures. Thus an analysis as the reviewer suggests is currently not possible with these models, although it would be worthwhile to study.

We assume that part of the objection seems to be due to the different definition of “effects of extreme temperatures”. In our case these comprise temperature-induced effects of water scarcity, photosynthesis etc. (see list in main paper introduction and above), while the reviewer seems to define them in a stricter way of damage to enzymes or flowering. This latter definition excludes indirect temperature-induced effects. But in the manuscript we make very clear that we refer to the wider definition. We use the process-based crop models to actually show that the observed yield reduction as response to exposure to high temperatures is likely due to temperature-induced water stress.

To analyze the temperature response without confounding factors we combine the strengths of process-based simulation models with a statistical model. As discussed in the manuscript we conclude that the negative yield response to high temperatures observed in both USDA and simulated yields is indeed due to water stress – in agreement with the last statement of the reviewer. We cannot and do not attempt to estimate temperature effects outside of the observed historical ranges where direct damage to enzymes may become relevant, i.e. at temperatures above approx. 40°C (therefore we also limit the future analysis to this temperature range).

Thus there is no new information on extreme heat stress provided by this study, in fact mere a wrong analysis of facts leading to wrong conclusions.

Response:

As described above, that point seems to be a matter mainly of the definition of “effects of high temperature” – which is wider in our case – and the aggregate scope of our study.

Furthermore, I strive to understand what use would a farmer have of the trivial facts discovered through this unnecessarily complex modeling machinery.

Response:

Process-based models are often used for projecting future crop yields under climate change and/or mitigation scenarios. Thus, we need to understand how reliable these models are. The fact that the process-based models capture the observed temperature responses make them highly relevant for outlining possible crop responses in the future. That temperature-induced yield losses can be mitigated to a large extent by irrigation may also be relevant for larger-scale adaptation planning. For the crop-modelling community the results are deemed to stimulate further improvement in representing temperature effects in individual models.

I would also suggest to avoid linking any effect on crop yield to food security considerations. Physiology and food security have very little in common and no logical link to each other (food security is determined by socio-economic and policy drivers well above and beyond crop function).

Response:

We fully agree with the reviewer that food security has several dimensions. Yet a sufficient provision of calories from agricultural production is a prerequisite for all of them^{15, 16} and is connected to yields on a spatially aggregated level, as we research in our study.

Reviewer #2 (Remarks to the Author)

The paper is an important contribution on an important topic. It's a paper that someone has needed to write for a long time -- so it's great that the authors have done so! -- and at least a subset of the results should be very useful to the broader debate about potential climate impacts on agriculture. I have one main concern, followed by some more minor stuff.

Response:

We want to thank the reviewer for the helpful, constructive and detailed comments. We adapted the manuscript accordingly.

1. There seems to be an important disconnect between what appears to be the main goal of the paper - to show that crop models and observations (statistical models) give comparable results - and a bunch of the other results they show which (1) either look only at crop model results, or (2) show results that are fairly inconclusive with regard to consistency between models and observations.

Response:

We apologize for not making the aim and structure of the manuscript more explicit. It actually consists of three consecutive steps described by the following questions:

1. Do models reproduce the observed decline in yields when crops are exposed to high temperatures during the growing season?

Answer: Yes, there is a quantitative agreement between model simulations and observations. The model ensemble reproduces the observed strong decline in yields at high temperatures under rainfed conditions. Model simulations and observations also agree in showing a much smaller decline (if at all) under irrigation.

2. What is the underlying mechanism of the yield decline?

Answer: In the model simulations the strong decline of yields at high temperatures and rainfed conditions is dominantly due to water stress, since the decline largely decreases when irrigation is “switched on”. Other factors can be ruled out based on different model results. The qualitative and quantitative agreement between the process-based model simulations and the observations shown in step 1 increases our confidence that the main driver of the simulated response is also the main driver of the observed response.

3. What do we expect for the future?

Answer: Model simulations allow for projections into a future under higher temperatures and higher levels of CO₂. Thus, in the final step we address the question whether the CO₂ fertilization effect, which is known to affect the water use efficiency of crops, may help to mitigate the strong decline in yields at high temperature levels. Although the CO₂ effect is expected to increase overall yields it is not expected to strongly reduce the temperature sensitivity of the crops – which may, on the contrary, be strongly reduced by irrigation. Again, the confidence in the temperature response of the models is based on the convincing results from step 1. It has to be noted, however, that temperature effects not relevant in the past may still become relevant in the future, and that a solid representation of temperature effects does not imply confidence in the representation of the CO₂ effect (see below).

We adapted the manuscript (in introduction, discussion and conclusion) to more clarity on our overall goal and conclusions.

For instance, the results section titled "Irrigation strongly attenuates the yield declines at heat (sic) temperatures" is somewhat confusing for at least 3 reasons: (1) the observed model for irrigated regions is very noisy (Fig 2a,b), meaning you can't really draw any conclusion here, (2) the crop models do suggest a clear reduction in yield above about 33C for corn and 30C for soy, and (3) the crop models do not appear to agree with each other (Fig S9b) about irrigated impacts at high temperatures.

Response:

We fully agree that the results for the irrigated areas (Figure 2 in main paper) are quite noisy. This may be due to, first, much less data for irrigated yields (2,277, 719, and 149 county-year entries for irrigated vs. 42,648, 41,920 and 38,845 for rainfed observed yields of maize, soybean and wheat, respectively) or, second, an actually weak temperature response of irrigated yields, in particular a lacking decline of yields at high temperature levels^{11, 12, 13, 14}. Thus, the analysis of the irrigated areas may provide a hint that the observed yield decline at high temperatures under rainfed conditions may be due to water stress or at least it does not speak against this hypothesis. However, solely based on this empirical finding it is definitely difficult to identify water stress as the main driver of the observed decline in yields under rainfed conditions. In this situation the process-based crop models offer an additional possibility to address the question by artificially switching on irrigation in the rainfed areas to see whether the decline lowers under this assumption. That is the essential advantage of the process-based models. The application of crop models to this question is therefore shown in Figure 1 of the main paper where we analyze irrigated yields in rainfed counties (blue lines).

We agree with the reviewer that there is a reduction in simulated maize and soybean yields with high temperatures even under irrigated conditions. But this decline is less pronounced and less clear than in the rainfed case, and it starts only at higher temperatures. The response of simulated irrigated maize and soybean yields in *rainfed* counties (blue lines in Figure 1) shows a similar pattern: almost no response to temperature except a decline at very high temperatures above 39°C (maize) or 36°C (soybean; only weakly). So we think that it is sound to conclude that irrigation strongly reduces the decline of crop yields at higher temperature although it may not disappear completely.

We also agree with the reviewer that there is a large spread between crop models with regard to the quantitative yield reduction under high temperatures for irrigated maize (former Figures S9b and S4a, now S10b and S5a). The decline of the median irrigated yield is driven by three models, EPIC-Boku, pAPSIM and pDSSAT. But even these show a later peaking (around 35°C) in the irrigated case and also a less pronounced decline when compared to the rainfed yields (Figures S10a and S4a). Note also that Figures S10-12 are for rainfed counties; this has not been stated clearly in the original supplement but is now added.

The models show a strong reduction of the decline in yields at high temperature levels as soon as irrigation is “switched on”, which supports the hypothesis that the decline is dominantly due to water stress. That is the main point we would like to make here, and the manuscript has been amended accordingly in the results and discussion section.

If we have to be cautious about the historical irrigated area results, then we also have to be cautious about the projections using irrigated area. Again, if the article is about comparing statistical models to crop models, then you can't conclude that "Irrigation can overcome detrimental heat effects also in the future" (next section header), because the irrigated results for the statistical model are presumably noisy enough to admit really negative estimates as well as really positive ones. So the proper summary of the result here would seem to be: "Some crop models suggest that irrigation can overcome about half of the negative impacts for rainfed areas, but estimates from statistical models are too noisy to draw this conclusion".

Response: We agree with the reviewer that we should not base any future projections on the rather noisy results for the irrigated areas. We want to thank the reviewer for highlighting that point. Yet the future results in the manuscript (Figure 3), where we test the interactions between CO₂, temperature and water, are only provided for the rainfed areas (see maps in Fig. 1) where we have a higher confidence in the crop models' simulation capacity. We have emphasized that point in the manuscript (results, discussion and Figure caption) and further clarified the hypothetical character of our model results.

Similarly on CO₂: we do not have an observational/statistical estimate of the fertilization effect (clearly very hard to get one, although people are trying). So again, are we just now supposed to believe the crop models on CO₂ because they got the rainfed effect "right", or should the authors again be clear that the CO₂ effect is from the crop models alone and thus should not be interpreted as something we know from observations. If it's the former, then the authors need to make this assumption very explicit - although it's hard to see how consistency on rainfed response under fixed CO₂ would convince us that the models get the CO₂ effect right.

Response: We apologize for not sufficiently highlighting that the CO₂ effect is taken from the models. We fully agree with the reviewer that an adequate sensitivity to temperature or water supply does not imply any conclusions on the adequacy of the CO₂ effect in models and have included that point in the discussion.

The models currently provide “best estimates” regarding the effects with model specific implementation varying across a wide range (see ref. 17 for a recent summary). As the models largely agree in showing only minor effects of elevated CO₂ concentrations on the yield decline at

high temperature (Figures S22-24), we assume there to be some robustness in this finding across the different implementations of the CO2 effect. But we have now further highlighted that our results are only hypotheses that may merit further investigation.

More minor:

2. Is correspondence between statistical models and crop models a surprise, or have the latter been calibrated to the former? A little more detail on that would be useful.

Response:

There has been no calibration of the models to fit the regression results presented in the manuscript (added this point to the Methods), in particular since this analysis has only been planned after the crop model results were submitted to the AgMIP repository. So we were indeed a bit surprised to see the high consistency, especially when bearing in mind previous caveats about the capacity of crop models to capture high temperature effects.

3. Fig S6: this figure seems important but it's not clear what's plotted here. these are the coefficients from all the different temperature bins?

Response:

Yes, Figures S7-9 (former S6-8) contain a 1:1 plot for each crop x irrigation setting in rainfed counties, showing the correspondence between regression coefficients from observed rainfed yields (x-axis) and simulated yields (y-axis; rainfed in *a* and irrigated in *b*). We have included these correlation plots for lack of an adequate statistical ensemble test that could tell whether the deviance of observed and simulated coefficients is statistically significant for all coefficients *at once* (i.e. similar to an F-test). Apart from inspecting confidence intervals of the single coefficients (as done in the main paper) we deduce from the visual proximity of the median correlation line and the 1:1 line for rainfed maize and soybean that the coefficients are sufficiently similar. For the irrigated simulation case (*b*), in contrast, the ensemble response visually diverges much more from the 1:1 line (maize and soybean). See also comments for wheat (Figure S9) below. We are sorry for the unclear descriptions in the supplement and adapted them.

4. Table S3: could get high R2 out of Fig S6 but sign could be wrong (e.g. see model plotted in blue). Maybe be clear about this? What are we to conclude from Fig S8?

Response:

We agree that the R2 values are rather high for maize and soybean in almost all cases, though the slopes differ. The R2 values are thus more or less inconclusive and we decided to omit this table. Figure S9 (former S8) shows the coefficient correlations for wheat, where there is no detectable difference between simulated rainfed and irrigated correlations with rainfed observed yields. The coefficients cluster around 0 (no detectable temperature response in simulated or observed yields), such that even negative slopes can occur spuriously. This plot thus confirms our conclusion about a missing high temperature signal for historic wheat yields. We added this explanation to the supplement.

5. Fig S9. caption appears to be wrong? panel b appears to show simulated rainfed (ensemble average is dotted line?) vs simulated irrigated, but caption says "rainfed observed versus irrigated simulated". or am I misunderstanding? the rainfed observed in panel a does not look like rainfed observed in panel b.

Response:

We apologize for the wrong caption and have adjusted both caption and selection of colors (now Figure 10). Panel b indeed shows the ensemble simulations under rainfed conditions for comparison, so the two black lines in the uncorrected supplement are not expected to look similar (there are no observed yields in panel b; the ensemble without irrigation is now grey).

Reviewer #3 (Remarks to the Author)

This paper describes an analysis of observed data and a process-based crop model ensemble to (1) determine if such crop models can capture observed heat effects, and (2) to simulate future crop yields with projected future climates. The authors demonstrate that the crop model ensemble can reproduce observed heat responses, and that these models project declined yields in the future. The paper is clearly written, and the references are appropriate.

Response:

We want to thank the reviewer for the helpful, constructive and detailed comments. We adapted the manuscript accordingly.

Previous work has established that high temperatures are detrimental to crop yields, and that climate change is likely to bring higher temperatures. Previous studies have also shown that crop models simulated with future climate scenarios exhibit reduced yields. However, this paper is of interest to the crop modelling community due to (1) their analysis of a curated ensemble of process-based climate models, (2) their investigation of the salient heat and CO₂ processes captured (or not) by this ensemble, and (3) their model-based argument that temperature-induced water stress is the key process in heat stress.

There are two issues that should be dealt with before publication.

First, it should be made clear in the main paper that, being restricted by available data, the ensemble is comprised of a small number of distinct models, particularly when projecting future wheat yields. Page 1 of the SI states that these 9 models "have distinct histories of development and are therefore individual with respect to implementation of temperature and CO₂ responses." The GEPIC, EPIC-Boku, and EPIC-IIASA models are all of the same model family, as are LPJ-GUESS and LPJml. Since ORCHIDEE-crop and pAPSIM were excluded in the future yield projections, there were 4 distinct model families used for future maize and soybean simulations, and 3 for future wheat simulations. The reader needs to be aware of this when interpreting the medians and 95% confidence intervals in the main figures, especially given the differences between individual models at high temperatures (e.g., figure 2 vs figure S14), and given that the maize model with the strongest negative irrigated response at high temperatures (pAPSIM, fig. S18) was excluded from future projections.

Response:

The reviewer raises an important point and is fully right that the models share common histories. Interestingly, though, even for the very same model core (as for the EPIC models) the yield response to environmental conditions (e.g. high temperatures) can strongly diverge due to different input and parameter choices that have not been harmonized for the considered AgMIP ensemble. It has been shown that the divergence between the three EPIC model versions in our ensemble can be as large as the divergence between models from different families. There is a publication on this topic underway (C. Folberth *et al.*) but unfortunately not yet available. There is a similar divergence in yield responses to environmental conditions for the LPJ family (see, for example, the different curves in Figures S4, S6 or S10), such that we would not necessarily expect a similar response of

individual models only because of the assignment to the same model family. Also, the LPJ-based models are actually quite distinct in how crop growth is implemented, including the phenology scheme as well as the allocation of NPP and associated feedback to productivity e.g. via leaf area¹⁸. We have included a paragraph on this topic into the discussion section and the supplement. Regarding the model ensembles used for future projections we agree with the reviewer that pAPSIM or ORCHIDEE-crop would have been interesting to include. But these models did not provide yield simulations under the ISI-MIP Fast Track protocol¹⁹ where the future yields were taken from. Therefore we had to resort to the six available models (five for wheat since PEGASUS does not simulate winter wheat), but did not deliberately exclude models from the ensemble.

Second, more work needs to be done to justify that the 0.5 degree spatial resolution can be used with their statistical method. Aggregating weather data can reduce local extremes, which are important with respect to crop heat stress. Perhaps this is why the interquartile ranges in figure S2 are all smaller for predicted than observed at high temperatures, for all crops? As the authors note, the AgMERRA data is an order of magnitude coarser than the data used by Schlenker and Roberts (S&R), and they are right to test the sensitivity of their coefficients to the climate data resolution. However, they do this by checking if the confidence intervals of their coefficients overlap with those of S&R. If confidence intervals do not overlap, there is statistical significance, but the opposite is not true.

Response:

We agree with the reviewer that extremes in temperature or precipitation could be flattened or lost in an aggregation procedure. To check whether the exposure to high temperatures is potentially underrepresented in the coarser AgMERRA data we have now included an additional comparison (SI Figure S2) in the supplement. It correlates the cumulative number of hot days (maximum temperature above either 30°C or 32°C) during all growing seasons within the analysis region for both climate data sets. These agree closely with R²=94% and 91%, respectively, where the AgMERRA data tend to include even more hot days than the fine-scale climate data in very hot regions.

In addition, we adjusted the text related to Figure S1 of the supplement stating that “there is no hint for a significant divergence of the regression coefficients based on the higher resolution temperatures and the ones based on the AgMERRA data”.

The slightly smaller IQRs in Figure S3 (former S2) are most likely due to the fact that the statistical model does not explain the full variance of the observed data. So the variance of the observational data is enlarged by the variance not explained by the statistical model. The use of aggregated climate data could contribute to a lower predicted variance, too, but given the tests above we assume this to be of minor relevance.

-- *Minor points* -

Perhaps justify why an ensemble of crop models is required for the future yield analysis, but only a single climate model is necessary (HadGEM2). If HadGEM2 is best for US, why not weight crop models according to how well they match observations?

Response:

We used one climate model in the analysis since we wanted to focus on the temperature response in crop models, rather than future projections of (absolute) yields. We expect the temperature response of the crop models – as analyzed with the panel regression – to be essentially independent from the climate model. In contrast, absolute yield projections may strongly differ from GCM to GCM due to different temperature and precipitation projections. A skill-weighted ensemble would also be worthwhile to inspect, but since the model versions between past and future are slightly different, we did not pursue this option in this manuscript. Our study can be considered as a demonstration of climate change effects, rather than an explicit quantification – which would require different GCMs

or emission scenarios. A sentence has been added to the Methods section to highlight this fact.

Replace 'most's with counts or percents (e.g., page 8: "Most models include...", page S5 "Individual models mostly ...", page 25 ".. and mostly even stronger so...").

Response:

We thank the reviewer for detecting these weak spots and improved the manuscript.

It would be useful to the reader to include an explanation of why ORCHIDEE-crop, EPIC-Boku and pDSSAT can deviate from observed so much (figure S3). Same goes for figure S14.

Response:

The most likely reason for the unexpected response is a low average yield for ORCHIDEE-crop, EPIC-Boku and LPJ-GUESS. ORCHIDEE-crop simulates only between 34-68% of the ensemble mean yields for all three crops, LPJ-GUESS simulates 51-68% of mean yields for maize and soybean (but 117% for wheat) and EPIC-Boku simulates 67% of mean yields for wheat. These are exactly the combinations where the models fail to follow the general temperature response pattern or grossly overestimate some coefficients. The low average yields seem to reduce the signal-to-noise ratio through an increased coefficient of variation, which results in an unexpected temperature response.

Regarding pDSSAT we assume that the reviewer refers to the spurious positive response at high temperatures for wheat in Figure S4 (former S3) and to the overall negative response for irrigated wheat in Figure S15 (former S14). This is most likely due to very few data in these cases, especially for irrigated wheat. The different response of pDSSAT for irrigated wheat in comparison with the other models would indeed require more research, though that is not a focus of this study and should be reproduced with more data before.

We added both interpretations to the supplement at the respective places.

'EPIC' is used as a model description throughout the SI (e.g., p. 21). Is it related to one of the 9 models in table S1?

Response:

The EPIC model used for future simulation is actually the EPIC-Boku model used for the historic analysis. The suffix "Boku" was only introduced when more models based on the same EPIC core joined the AgMIP project. We are sorry for the confusion and have replaced "EPIC" by "EPIC-Boku" in the manuscript.

References

1. Lobell DB, Hammer GL, McLean G, Messina C, Roberts MJ, Schlenker W. The critical role of extreme heat for maize production in the United States. *Nature Climate Change* **3**, 497-501 (2013).
2. Deryng D, Conway D, Ramankutty N, Price J, Warren R. Global crop yield response to extreme heat stress under multiple climate change futures. *Environmental Research Letters* **9**, 034011 (2014).
3. Barlow KM, Christy BP, O’Leary GJ, Riffkin PA, Nuttall JG. Simulating the impact of extreme heat and frost events on wheat crop production: A review. *Field Crops Research* **171**, 109-119 (2015).
4. Barnabas B, Jager K, Feher A. The effect of drought and heat stress on reproductive processes in cereals. *Plant, cell & environment* **31**, 11-38 (2008).
5. Lobell DB, Sibley A, Ivan Ortiz-Monasterio J. Extreme heat effects on wheat senescence in India. *Nature Climate Change* **2**, 186-189 (2012).
6. Wahid A, Gelani S, Ashraf M, Foolad M. Heat tolerance in plants: An overview. *Environmental and Experimental Botany* **61**, 199-223 (2007).
7. Grant RF, Kimball BA, Conley MM, White JW, Wall GW, Ottman MJ. Controlled Warming Effects on Wheat Growth and Yield: Field Measurements and Modeling. *Agronomy Journal* **103**, 1742-1754 (2011).
8. Schlenker W, Roberts MJ. Nonlinear temperature effects indicate severe damages to U.S. crop yields under climate change. *Proceedings of the National Academy of Sciences of the United States of America* **106**, 15594-15598 (2009).
9. Luo Q. Temperature thresholds and crop production: a review. *Climatic Change* **109**, 583-598 (2011).
10. Rötter RP, Carter TR, Olesen JE, Porter JR. Crop–climate models need an overhaul. *Nature Climate Change* **1**, 175-177 (2011).
11. Troy TJ, Kipgen C, Pal I. The impact of climate extremes and irrigation on US crop yields. *Environmental Research Letters* **10**, 054013 (2015).
12. Hawkins E, Fricker TE, Challinor AJ, Ferro CAT, Ho CK, Osborne TM. Increasing influence of heat stress on French maize yields from the 1960s to the 2030s. *Global change biology* **19**, 937-947 (2013).
13. Tack J, Barkley A, Nalley LL. Effect of warming temperatures on US wheat yields. *Proceedings of the National Academy of Sciences of the United States of America* **112**, 6931-6936 (2015).
14. Siebert S, Ewert F, Eyshi Rezaei E, Kage H, Graß R. Impact of heat stress on crop yield—on the importance of considering canopy temperature. *Environmental Research Letters* **9**, 044012 (2014).

15. Schmidhuber J, Tubiello FN. Global food security under climate change. *Proceedings of the National Academy of Sciences of the United States of America* **104**, 19703-19708 (2007).
16. van Ittersum MK, Cassman KG, Grassini P, Wolf J, Tittone P, Hochman Z. Yield gap analysis with local to global relevance—A review. *Field Crops Research* **143**, 4-17 (2013).
17. Deryng D, *et al.* Regional disparities in the beneficial effects of rising CO₂ concentrations on crop water productivity. *Nature Climate Change*, (2016).
18. Lindeskog M, *et al.* Implications of accounting for land use in simulations of ecosystem carbon cycling in Africa. *Earth System Dynamics* **4**, 385-407 (2013).
19. Warszawski L, Frieler K, Huber V, Piontek F, Serdeczny O, Schewe J. The Inter-Sectoral Impact Model Intercomparison Project (ISI-MIP): project framework. *Proceedings of the National Academy of Sciences of the United States of America* **111**, 3228-3232 (2014).

Reviewers' comments:

Reviewer #1 (Remarks to the Author):

I read the replies of the authors and I think that they continue to misunderstand the critical points I had made in my first review.

Changing the name 'heat' with 'high temperature' is more correct, but it does not really solve the problems I had highlighted earlier. The real clarification needed would be between high temperature and extremely high temperature events. The former happen regularly during most growing seasons, however the latter are extreme events that are difficult to see -and thus test against--when using large spatial and temporal averages.

It is the extreme events info that is missing in crop models. Talking about high temperatures is a non issue.

(As a scientist involved in crop modeling for many years, I am well aware that the models used do not respond to temperature anomalies but run on actual temperature, including min max values and sometimes all values I between. Most of the modes employed in this study, to be precise, run on daily mean temperature.)

The models used in the manuscript contain a number of equations that describe (rather well) the direct impact of high temperature on physiological processes such as photosynthesis and transpiration. However, the models used do not employ any equation that captures physical damage to extreme high temperatures. This is in a nutshell the same point made in the existing literature wet limitations of current crop models in capturing extremely high temperature effects. And because climate change will bring with it increased frequencies of extreme events, everyone expects that current crop models will need some modifications to stay relevant under those changed climate regimes .

With regards to the above, every single paper I know that attempts to improve on the ability of crop modes in this area, first develops updated models with new equations that capture more processes relevant to extreme temperatures, and then tests against observations to show improved agreement with specific observations.

But the authors of this manuscript have not done that. Rather, they took existing off the shelves models and ran analysis against observations to show that they do a good job against high temperatures (but never look at extremely high temperatures).

So their conclusion seems to be that the crop models used, despite previous statements to the contrary, can indeed capture effects of high temperatures, and that these are Largely mediated by water interactions. And this conclusion is as true as irrelevant, given the above.

In reality, What the authors have looked at is something other than effects of extremely high temperature. they have carried out an analysis where the yields signals in question are averaged over space (county level) and time (several years averages). These averages are precisely those that would tend to mask any local effect, in time and space, of extremely high temperatures. So what is left, and what the authors indeed have found, is precisely what these models Are already known in the agronomic literature to be able to Capture well: impacts of high temperature on phenology, water interactions, etc. there is no news in showing that this agrees with the historical record: it has been done countless times before, except that the authors are now, despite acknowledging in their response that they only deal with high temperatures, framing the exercise in a way that seems to be particularly relevant for new climate change findings worth of publication in nature. In my opinion it is not, since what we need to know are the reposes to extremely high temperatures under climate change.

On the point about irrigation responses and leaf temperature, I do not agree with the view of the

authors that because of transpiration the response under an irrigation regime would be lower... The issue at hand is one of with and w/o climate change, and leaf temperature would work similarly under current and future climate conditions.

On the point on food security, of course it all depends on the initial calorie provision. However my point is that we currently have enough calorie provision and yet still nearly a billion people hungry. The infrastructure and policy dimensions to this are so much more important than the underlying agronomy that making that link is fashionable but misleading.

Reviewer #2 (Remarks to the Author):

I think the authors had addressed my concerns adequately, and I will not trouble them further. I think this paper will be useful to the community.

Reviewer #3 (Remarks to the Author):

The updated manuscript is significantly improved and has sufficiently addressed the issues raised in my review, with the exception of spatial aggregation (detailed in next paragraph). Since this study is focused on the consistent response found between the model ensemble and observations, rather than projections of actual yields, the aggregation issue does not diminish their main findings or conclusions. But given the wide readership and high profile of this journal, I strongly encourage the authors to at least acknowledge aggregation as a potential issue in the main text (e.g., replacing lines 379-380 of the revised manuscript, and lines 67-68 of the supplementary material, which are currently misleading).

Spatial aggregation is an important issue when analysing extremes. The authors take the right approach to assessing the disparity between the fine scale and coarse scale data by checking whether high temperatures are underrepresented in the coarser dataset. But they compare the coarser dataset with a spatially aggregated version of the fine-scale dataset, i.e. they averaged the fine-scale data to the 0.5 degree scale, found this comparable with the coarse 0.5 degree data, and then use this comparison to claim that the coarser data is representative of finer-scale extremes. It's the effects of aggregation, such as this averaging, that are the issue here, as extremes can be removed in the process. A more representative comparison would be between the normalized frequency distributions of the two datasets at their native spatial resolutions.

Response to Reviewers – Round 2

We want to thank the three reviewers for commenting on our manuscript in a second round. We appreciate their remarks and would like to provide responses to the individual points below. Based on the very helpful suggestions by the reviewers we have substantially expanded the analysis and revised the manuscript. First, we included a sensitivity test whether the applied regression approach is able to detect artificially included yield losses in the extremely high temperature range (above 36°C). Second, a quantification of the yield losses expected solely by increased exposure to temperatures in the range from 30°C to 36°C is added. This underlines the relevance of yield losses induced by longer exposure to “high” temperatures under climate change, independent from effects of “extremely high” temperatures. Third, we provide additional information about the timing of exposure to “extremely high” temperatures within the growing season to show that the estimated effect of these temperatures does not appear reduced just because critical temperatures occurred only in insensitive phases of the growing period. Finally, we provide an additional comparison of the temperature distributions used in our study (based on AgMERRA) and the higher resolution data applied in the study by Schlenker & Roberts, as requested by the third reviewer.

We provide two versions for both manuscript and supplement: one with all changes to the previous version marked to facilitate their tracking and one where these changes are flattened into one document.

Reviewer #1 (Remarks to the Author):

I read the replies of the authors and I think that they continue to misunderstand the critical points I had made in my first review.

Response: We would like to thank the reviewer for a second inspection of our manuscript. Before addressing the individual concerns we would like to stress that none of them calls the scientific soundness of our analysis into question. Instead there seems to be a different understanding of the focus and relevance of our study. We directly respond to the individual issues below and provide a more general response at the end. Wherever possible we adapted the manuscript to more clarity and extended the discussion including the reviewer’s concerns.

Changing the name 'heat' with 'high temperature' is more correct, but it does not really solve the problems I had highlighted earlier. The real clarification needed would be between high temperature and extremely high temperature events. The former happen regularly during most growing seasons, however the latter are extreme events that are difficult to see -and thus test against--when using large spatial and temporal averages.

Response: To straighten the discussion we define ‘high temperatures’ as temperatures above 30°C and ‘extremely high temperatures’ as temperatures above 36°C. This is a consistent intermediate value from previous studies^{1, 2, 3, 4, 5, 6} for the threshold above which direct temperature-induced damage to enzymes, tissues or reproductive organs of the plant is expected. It is important to note

that we calculate exposure times at each grid cell individually and that these exposure times are then averaged spatially to counties. We do not average the temperatures themselves. Thus no exposure to high or extremely high temperatures at one of the grid cells is lost. Averaging certainly produces smaller county average exposure times compared to individual exposure times if these are singular events across the region. However, that kind of reduction only reflects the expected reduced influence of this singular event on the reported county average crop yield. County-level mean exposure times reach 10.8, 13.1 and 6.0 days above 30°C per average growing season in the historic climate data set for maize, soybeans and wheat, respectively. Above 36°C there are 0.7, 1.1 and 0.5 days per average growing season (same crop order). Annual county-level mean exposure to extremely high temperatures adds up to a total of 41,580, 70,934 and 34,200 days with maximum temperature above 36°C in our panel, across all years and counties for maize, soybean and wheat. Exposure times for soybean are larger than for maize, despite the equal growing season and a similar number of county-year entries in the panel, since its growing area extends further to the south. We consider this a sufficient amount for our panel analysis, and a unique potential of the panel regression spanning the large area compared to sparse field data.

To illustrate the sensitivity of the regression to potential yield losses from extremely high temperatures up to 42°C we made use of the yield simulations at 0.5° grid cell level. These allow for testing whether disturbances at that level will be seen in the panel regression after aggregation to county level. To this end simulated yields at each grid cell were artificially reduced by 2% for each day spent at temperatures above 33/36/39°C (three separate analyses). Afterwards the grid-level data were aggregated to county level and evaluated by the panel regression as described in the manuscript. The regression shows a clear and quantitatively correct signal in the temperature coefficients where the reductions were applied, while the others stay unaffected (Figure S5 now included in the Supplement). We conclude that there is a sufficient amount of ‘extreme’ temperature events in the past for our analysis, and that the statistical model applied is sensitive to these.

We agree that even more extreme temperatures and associated yield responses may have occurred on individual fields that are not resolved by the statistical approach. However, the analysis based on higher resolved temperatures shows that the coefficients are not sensitive to this increase in resolution (section 2 of the SI, Figure S3). The comparison based on county level of aggregation is, on the one hand, constrained by the available data but, on the other hand, also represents a meaningful level of aggregation for the global gridded crop models that are typically forced by weather data on 0.5°x0.5° resolution⁷ and often applied to simulate yield changes for even larger regions⁸. Even though crop models have been shown to respond well to additional exposure to high temperatures on individual field scale after calibration, this has not been shown in such detail at larger spatial scales as we do here, and the good agreement with observations adds substantial trust to the models that are largely uncalibrated at these scales.

It is the extreme events info that is missing in crop models. Talking about high temperatures is a non issue.

(As a scientist involved in crop modeling for many years, I am well aware that the models used do not respond to temperature anomalies but run on actual temperature, including min max values and

sometimes all values I between. Most of the models employed in this study, to be precise, run on daily mean temperature.)

The models used in the manuscript contain a number of equations that describe (rather well) the direct impact of high temperature on physiological processes such as photosynthesis and transpiration. However, the models used do not employ any equation that captures physical damage to extreme high temperatures. This is in a nutshell the same point made in the existing literature wet limitations of current crop models in capturing extremely high temperature effects. And because climate change will bring with it increased frequencies of extreme events, everyone expects that current crop models will need some modifications to stay relevant under those changed climate regimes.

Response:

We fully agree with the reviewer that the models in our ensemble (except PEGASUS, Table S1) contain no direct damages from extreme temperatures. That is consensus. However, they are nonetheless able to capture the response curve of the observed yields even quantitatively. We added a further discussion about this fact to the manuscript, which is copied here:

“The statistical approach is sensitive to yield losses induced by extremely high temperatures, despite their low relative abundance in the data set (SI Figure S5). At the same time the direct damage to enzymes, tissues or reproductive organs expected in these temperature ranges is not represented in the crop models (see above). Thus, the agreement between observations and simulations indicates that damage directly induced by extremely high temperatures is of minor relevance in the historical sample on the spatial scale of our study.

Damages in the observed yields could in fact be limited if temperatures occurred in noncritical periods of the growing season, but in the considered sample extreme temperatures mainly occurred in the middle and last phase of the growing season, in which anthesis and grain filling mostly occur (SI Figure S6). Both these processes are known to be critically sensitive to high temperatures. In addition, a sensitivity test regarding the timing of the exposure and the definition of the growing season has not revealed any significant difference in the associated responses to extreme temperatures⁹.

In addition, evaporative cooling may have reduced leaf temperatures to lower values than air temperatures, which are used as predictor in the regression model. The latter aspect is not represented in the crop models and requires further work to quantify the role of evaporative cooling as a protection mechanism^{10, 11}. In addition, harvests may have been adjusted to avoid exposure to extremely high temperatures, an effect not represented in the exposure times used in our analysis. Yet, given the abundant total number of such extremely high temperatures in our data set (41,580 days above 36°C for maize, 70,934 for soybean and 34,200 for wheat) we assume that the latter explanation is less relevant.

The agreement between the observed and simulated temperature sensitivities found for the historical sample does not imply that models capture all processes relevant under future climate change where direct temperature-induced damages may become more relevant. However, based on the regression coefficients derived from the historical observations and temperature shifts projected for the end of the century by HadGEM2-ES under RCP8.5, increasing exposure to temperatures in the range from 30°C to 36°C alone implies yield losses of 49% for maize, 40% for soybean, and 22% for wheat (Table

1). *Our analysis suggests that crop models reliably simulate temperature effects in this range. A further test of the reliability of future projections of yield losses could be achieved by assessing regions that are already warmer today or of field experiments where temperatures are artificially increased^{12, 13}.*”

We do not deny anywhere the necessity to adapt crop models to account for direct effects of extreme temperatures and have stated this more clearly in the discussion:

“Though extreme temperatures will become more important under climate change, and crop models will have to capture the associated effects¹⁴ [...]”,

“Direct crop damages from extremely high temperatures (e.g. 40°C) are usually not represented in current crop models and would have to be improved before assessing crop responses to these extremes in the future¹⁴.”

and

“The agreement between the observed and simulated temperature sensitivities found for the historical sample does not imply that models capture all processes relevant under future climate change where direct temperature-induced damages may become more relevant.”

Notwithstanding, we disagree with the reviewer’s opinion that climate change impact studies with existing crop models are obsolete as long as they do not include an improved response to extremely high temperatures. Already an extended exposure to temperatures in the range from 30°C to 36°C is expected to lead to substantial crop yield losses, since temperatures beneficial for yield formation are replaced by ones that are detrimental. We have quantified these effects and added them to the manuscript (see text above and Table 1 in the main paper).

With regards to the above, every single paper I know that attempts to improve on the ability of crop models in this area, first develops updated models with new equations that capture more processes relevant to extreme temperatures, and then tests against observations to show improved agreement with specific observations.

But the authors of this manuscript have not done that. Rather, they took existing off the shelves models and ran analysis against observations to show that they do a good job against high temperatures (but never look at extremely high temperatures).

Response: On these points we would like to refer to our general response below. Our focus is not on model improvement, but on model assessment, mechanism detection and interactions with climate change or CO₂. We are able to demonstrate, firstly, good model skill in reproducing observed yield responses at larger scales, secondly, that changes in exposure to high temperatures under climate change cause substantial damage to crop yields which, thirdly, can be mitigated by irrigation, but, fourthly, likely not by increased CO₂. We acknowledge that the detrimental effects of exposure to extremely high temperatures may not be visible in current county-level yield statistics as these events are currently too rare and that most current crop models do not account for the relevant mechanisms of very high temperatures; please see also our response above.

So their conclusion seems to be that the crop models used, despite previous statements to the contrary, can indeed capture effects of high temperatures, and that these are Largely mediated by water interactions. And this conclusion is as true as irrelevant, given the above.

Response: On the ‘irrelevant’ results we would like to point to our additional analysis where it becomes obvious that also the temperature response in non-extreme bins is of utmost relevance for climate change impacts. We also want to note that our study is the first-of-a-kind to separately *quantify* model responses to temperature exposure within individual bins, which are otherwise only researched on field or experimental scale.

In reality, What the authors have looked at is something other than effects of extremely high temperature. they have carried out an analysis where the yields signals in question are averaged over space (county level) and time (several years averages).

These averages are precisely those that would tend to mask any local effect, in time and space, of extremely high temperatures.

Response: We estimate the average *response* to times spent in individual temperature bins (but not to average *temperatures*). We very explicitly describe that our analysis covers the full range of temperatures above 0°C and that we estimate the response to exposure times for all 3°C bins covering the entire range. To further highlight the kind of response we estimate and the temperature range we consider we have added the term “average” several times when describing yield responses to average exposure times within each county, e.g. at the following places:

“We show that an ensemble of nine crop models reproduces the observed average temperature responses of US maize, soybean and wheat yields.” (in the abstract)

and

“The considered ensemble of nine GGCMs (eight for wheat) is able to closely reproduce the observed average response of rainfed crop yields (γ_h , see Material & Methods equation (1)) to time spent in different temperatures from 0 to 42°C (Figure 1, green and black lines).” (in the results section)

So what is left, and what the authors indeed have found, is precisely what these models Are already known in the agronomic literature to be able to Capture well: impacts of high temperature on phenology, water interactions, etc. there is no news in showing that this agrees with the historical record: it has been done countless times before, except that the authors are now, despite acknowledging in their response that they only deal with high temperatures, framing the exercise in a way that seems to be particularly relevant for new climate change findings worth of publication in nature. In my opinion it is not, since what we need to know are the reposes to extremely high temperatures under climate change.

Response: Once again we would like to point to our response above that not only the response to extreme temperatures is relevant. The reviewer’s claim that our analysis ‘has been done countless times before’ is, to our best knowledge, not true, given the scale of application, the detailed

comparison to observation and the application of a crop model ensemble to examine irrigation as an adaptation measure to high temperature exposure. There is no study that evaluates crop models with respect to responses to individual temperature exposure (which is different from mean growing season temperatures, which have indeed been evaluated, e.g. ref¹⁵), and also no study that shows these models capturing the observed cut-off (non-linear) behavior. Both are relevant to know for climate change impacts. See also our general response below.

On the point about irrigation responses and leaf temperature, I do not agree with the view of the authors that because of transpiration the response under an irrigation regime would be lower... The issue at hand is one of with and w/o climate change, and leaf temperature would work similarly under current and future climate conditions.

Response: We fully agree with the reviewer that the relevant temperature-related mechanisms (damage, water stress, temperature regulation etc.) are expected to work similarly in past and future for the same temperature. Thus, modeling leaf or canopy temperature is one major avenue for improving crop response to ambient air temperature, water and CO₂ (or ozone etc.) interactions¹¹.

Regarding the effects of heat stress under irrigation we would like to refer to our response and the cited references in the first 'Response to Reviewers' document (page 4), to which we have little to add. The relevant part is repeated here:

“We agree with the reviewer that when only considering direct physiological effects of high temperature the crop response may be more clearly visible under irrigation, when variations due to water stress are excluded. Yet [...] this is not the focus of our study. There is also a manifold of articles showing that irrigation (or ample water supply) buffers crops from high temperature stress^{11, 16, 17, 18}. When excluding water stress and looking only at direct physiological damages one would have to consider leaf temperatures, which are lower for irrigated yields (more water allows more transpirational cooling), possibly leading to less damage (and not larger ones, as the reviewer states).”

We explicitly discuss the issue of direct damages, potentially induced by temperatures reaching the highest two bins considered in our study, in comparison to the effects translated by water stress already occurring at lower temperatures (see excerpt of the revised manuscript in the third response above).

On the point on food security, of course it all depends on the initial calorie provision. However my point is that we currently have enough calorie provision and yet still nearly a billion people hungry. The infrastructure and policy dimensions to this are so much more important than the underlying agronomy that making that link is fashionable but misleading.

Response: We agree with the reviewer on the complex nature of food security and have removed the reference in our manuscript.

General response:

Individual yields, even if reported for one specific site, do not allow for an estimation of yield responses to specific temperatures. They only represent the integrated response to all weather fluctuations across the growing season. In principle there are two approaches to disaggregate the responses. The first is experimental. In this case temperatures are experimentally increased during a certain period of the growing season and the effect of this specific increase can be quantified by comparison to a control experiment without the perturbation (e.g. ^{12, 13, 15}). The second approach is empirical. It builds on a large sample of observations where the individual responses to exposure time can be estimated based on the variation of exposure times and yields across the sample.

The experimental approach has the clear advantage of maximum control over the temperature manipulation. It represents a very direct way of attribution. Yet such experiments are still very sparse and therefore limited to very specific conditions regarding cultivar, soil, management conditions, fluctuations in other variables such as available water or underlying undisturbed temperatures, the timing of the perturbations, *etc.* The majority of these isolated experimental results have not yet been incorporated into modeling simply because there are not enough data to develop robust functions needed for inclusion into models. Therefore it is difficult (and risky) to scale them up to larger geographic regions which is the main focus of this work. The panel regression approach (with fields or counties as the spatial base unit), represents a more indirect way of attributing yield fluctuations to times spent in different temperature intervals. In contrast to individual-site responses to specific perturbations the estimated coefficients represent average responses across a wide range of different growing conditions as they are assumed to be common to all elements in the pool. Our analysis is therefore complementary to field experiments and is possibly helpful for adaptation planning. It would certainly be appealing to expand the panel regression approach to field scale to avoid the spatial aggregation of exposure times and yield responses within counties. But the statistical approach requires a critical number of observations to abstract from possible reporting errors or confounding variables. While site-based observations are still sparse, the considered sample of county data provides sufficient statistical power as it builds on reported crop yields from several years (31) and a large number (2,982) of counties.

Any analysis of crop yield responses to high temperatures will have to be adjusted to the available data. Given the current data constraints, where controlled heat-disturbance experiments are only starting^{12, 13, 15}, our study comes as close as possible to the required disaggregation of the total yield response. The averaging over genotype x environment x management combinations^{10, 19} is inevitable when studying regions beyond field scale. But this also allows for the derivation of a more general response than the analysis of very specific field experiments.

We also consider it an important scientific contribution that we test the range of models that are usually applied for large-scale yield projections. Taking the models explicitly “off the shelves” means that our study does not focus on a new implementation of the direct physical damage of extremely high temperatures into a process-based crop model. Such type of work clearly needs to be done^{10, 14, 20}, but separately by each modeling group at plant/field scale (when there is enough data). Then new versions of the updated models will be released for application at regional scale. Thus it is necessary to use “off the shelves” models for regional scale analyses and to better understand how well these

perform. To our best knowledge all studies that assess or improve heat effect implementations are restricted to very few observation sites (e.g. ^{14, 21, 22}). But it is a different and equally relevant question whether the well-established models are able to reproduce the average response to temperature variations derived from a large scale region. The covered region is on the spatial scale the models considered here are designed to operate on. They provide input, for example, for global (economic) assessments addressing the risks of food supply shortages (e.g. ^{7, 8, 23}) and therefore have to be evaluated carefully. Our ensemble provides a broad coverage across this group of models.

Reviewer #2 (Remarks to the Author):

I think the authors had addressed my concerns adequately, and I will not trouble them further. I think this paper will be useful to the community.

Response:

We thank the reviewer for providing constructive comments in the first round and accepting our modifications based on these.

Reviewer #3 (Remarks to the Author):

The updated manuscript is significantly improved and has sufficiently addressed the issues raised in my review, with the exception of spatial aggregation (detailed in next paragraph). Since this study is focused on the consistent response found between the model ensemble and observations, rather than projections of actual yields, the aggregation issue does not diminish their main findings or conclusions. But given the wide readership and high profile of this journal, I strongly encourage the authors to at least acknowledge aggregation as a potential issue in the main text (e.g., replacing lines 379-380 of the revised manuscript, and lines 67-68 of the supplementary material, which are currently misleading).

Response:

We thank the reviewer for this helpful advice. We adjusted the relevant section of the main paper according to the suggestion (see extract below in quotation). We deleted the sentence in the supplement but added a further comparison (see below). We fully acknowledge spatial aggregation as a potential issue in such an analysis, though we are convinced that in our particular case this should only marginally increase uncertainty.

“We employed daily maximum and minimum temperature and precipitation data for the statistical model, and further weather variables for the yield simulations by the Global Gridded Crop Models

(GGCMs), from the AgMERRA climate data set, covering the years 1980 to 2010. The weather data were spatially aggregated to 0.5° for the crop simulations. We used the identical data set for the statistical analysis. Its spatial resolution is one order of magnitude coarser than in the original empirical study, which could result in less temperature extremes due to aggregation effects. But the slight deviation between the temperature distributions of the two data sets (SI Figures S1 and S2) only has a minor effect on the estimated coefficients (SI Figure S3). Additionally, predicted yields from the regression model based on the AgMERRA data are in close agreement with the observed yields in terms of mean growing season temperatures (SI Figure S4)."

Spatial aggregation is an important issue when analysing extremes. The authors take the right approach to assessing the disparity between the fine scale and coarse scale data by checking whether high temperatures are underrepresented in the coarser dataset. But they compare the coarser dataset with a spatially aggregated version of the fine-scale dataset, i.e. they averaged the fine-scale data to the 0.5 degree scale, found this comparable with the coarse 0.5 degree data, and then use this comparison to claim that the coarser data is representative of finer-scale extremes. It's the effects of aggregation, such as this averaging, that are the issue here, as extremes can be removed in the process. A more representative comparison would be between the normalized frequency distributions of the two datasets at their native spatial resolutions.

Response:

Figure 1 below shows the comparison of normalized temperature distributions from the two data sets at their resolution as applied in the regressions. The distribution of the fine-scale data originally used in the study by Schlenker & Roberts⁹ is slightly shifted to higher values. However, the effect on the estimated yield responses seems to be minor as shown in Figure S3 of the Supplement. We consider that the critical test for the issue of the spatial aggregation. It shows that potential differences between the observed responses derived in the original study by Schlenker & Roberts and the simulated yield responses cannot be explained by differences in the temperature data sets. As the crop models were forced by the AgMERRA data we consider it adequate to use this data set for our analysis. It ensures that the responses derived from the crop model simulations are based on exactly the same temperatures they receive as input – otherwise sources for differences would be difficult to disentangle. We have added the figure to the SI (Figure S1).

It is important to note that the fine-scale climate data used in Schlenker & Roberts are derived from monthly data with high spatial resolution (PRISM data set; see appendix of their study) and daily data from weather stations with an irregular spatial resolution. Although much care has been taken by Schlenker & Roberts to compile the highly resolved climate data set, it is expected to contain deviations from "true" temperatures on its own. Therefore comparisons between the fine-scale and the AgMERRA climate data show only differences between the two data sets, but not necessarily differences to the true climate.

Figure 1: Normalized frequency distribution of daily maximum temperatures as derived from the two observational climate data sets used in this study (yellow: temperature data used in the original study by Schlenker & Roberts with a spatial resolution of about $0.04^\circ \times 0.04^\circ$; blue: temperature data from the AgMERRA data set used in our study and applied to force the crop model simulations with a spatial resolution of $0.5^\circ \times 0.5^\circ$). The distributions are based on the sample of all daily maximum temperatures across all grid cells without spatial or temporal aggregation. No land-use weighting has been applied.

1. Barlow KM, Christy BP, O’Leary GJ, Riffkin PA, Nuttall JG. Simulating the impact of extreme heat and frost events on wheat crop production: A review. *Field Crops Research* **171**, 109-119 (2015).

2. Lobell DB, Sibley A, Ivan Ortiz-Monasterio J. Extreme heat effects on wheat senescence in India. *Nature Climate Change* **2**, 186-189 (2012).
3. Crafts-Brandner SJ, Salvucci ME. Sensitivity of photosynthesis in a C4 plant, maize, to heat stress. *Plant Physiol* **129**, 1773-1780 (2002).
4. Djanaguiraman M, Prasad PVV, Boyle DL, Schapaugh WT. Soybean Pollen Anatomy, Viability and Pod Set under High Temperature Stress. *Journal of Agronomy and Crop Science* **199**, 171-177 (2013).
5. Porter JR, Gawith M. Temperatures and the growth and development of wheat a review. *European Journal of Agronomy* **10**, 23-36 (1999).
6. Sanchez B, Rasmussen A, Porter JR. Temperatures and the growth and development of maize and rice: a review. *Global change biology* **20**, 408-417 (2014).
7. Rosenzweig C, et al. Assessing agricultural risks of climate change in the 21st century in a global gridded crop model intercomparison. *Proceedings of the National Academy of Sciences of the United States of America* **111**, 3268-3273 (2014).
8. Nelson GC, et al. Climate change effects on agriculture: economic responses to biophysical shocks. *Proceedings of the National Academy of Sciences of the United States of America* **111**, 3274-3279 (2014).
9. Schlenker W, Roberts MJ. Nonlinear temperature effects indicate severe damages to U.S. crop yields under climate change. *Proceedings of the National Academy of Sciences of the United States of America* **106**, 15594-15598 (2009).
10. Boote KJ, Jones JW, White JW, Asseng S, Lizaso JI. Putting mechanisms into crop production models. *Plant, cell & environment* **36**, 1658-1672 (2013).
11. Siebert S, Ewert F, Eyshi Rezaei E, Kage H, Graß R. Impact of heat stress on crop yield—on the importance of considering canopy temperature. *Environmental Research Letters* **9**, 044012 (2014).
12. Grant RF, Kimball BA, Conley MM, White JW, Wall GW, Ottman MJ. Controlled Warming Effects on Wheat Growth and Yield: Field Measurements and Modeling. *Agronomy Journal* **103**, 1742-1754 (2011).
13. Wall GW, Kimball BA, White JW, Ottman MJ. Gas exchange and water relations of spring wheat under full-season infrared warming. *Global change biology* **17**, 2113-2133 (2011).

14. Maiorano A, *et al.* Crop model improvement reduces the uncertainty of the response to temperature of multi-model ensembles. *Field Crops Research*, (2016).
15. Asseng S, *et al.* Rising temperatures reduce global wheat production. *Nature Climate Change* **5**, 143-147 (2014).
16. Hawkins E, Fricker TE, Challinor AJ, Ferro CAT, Ho CK, Osborne TM. Increasing influence of heat stress on French maize yields from the 1960s to the 2030s. *Global change biology* **19**, 937-947 (2013).
17. Tack J, Barkley A, Nalley LL. Effect of warming temperatures on US wheat yields. *Proceedings of the National Academy of Sciences of the United States of America* **112**, 6931-6936 (2015).
18. Troy TJ, Kipgen C, Pal I. The impact of climate extremes and irrigation on US crop yields. *Environmental Research Letters* **10**, 054013 (2015).
19. Messina CD, *et al.* Limited-Transpiration Trait May Increase Maize Drought Tolerance in the US Corn Belt. *Agronomy Journal* **107**, 1978 (2015).
20. Holzworth DP, *et al.* Agricultural production systems modelling and software: Current status and future prospects. *Environmental Modelling & Software* **72**, 276-286 (2015).
21. Eitzinger J, *et al.* Sensitivities of crop models to extreme weather conditions during flowering period demonstrated for maize and winter wheat in Austria. *Journal of Agricultural Science* **151**, 813-835 (2012).
22. Liu B, Asseng S, Liu L, Tang L, Cao W, Zhu Y. Testing the responses of four wheat crop models to heat stress at anthesis and grain filling. *Global change biology* **22**, 1890-1903 (2016).
23. Field CB, *et al.* Climate Change 2014: Impacts, Adaptation, and Vulnerability. Part A: Global and Sectoral Aspects. Contribution of Working Group II: Food security and food production systems. (ed[^](eds (IPCC) IPoCC). Cambridge University Press (2014).

Reviewers' Comments:

Reviewer #1 (Remarks to the Author)

I believe the manuscript can now be published.

I commend the authors for their additional revisions, which meet my earlier request for adding clarifications with respect to the specific dynamics of high temperature impacts on cereals, distinguishing between average processes and local events linked to extreme conditions.

I would like to see two more minor additions to the final manuscript:

1. Some text that clarifies that the average crop dynamics of high temperature as mediated by phenology and water are well understood and have so for decades--albeit this paper undoubtedly adds significant numerical detail to tease which specific processes may have been determinant to yield over the historical period observed; and

2. Some text that mentions farmer adaptation under climate change. Again, it has been very well recognized and for decades that higher temperatures connected to climate change would have negative impacts on yields--sometimes even considerably. However, it has also been amply recognized that reporting negative yields by X% needs additional qualifiers...meaning they do not consider adaptation. In reality, before large impacts could set in, farmers would have shifted to cultivars better adapted to the new temperature regimes --within possibilities.

So some statement mentioning that the reported climate change results are valid without adaptation --would be more precise. Of course, the authors may further speculate on the fact that, adapted or not--many cultivars would nonetheless suffer in future decades from an increased frequency of extreme temperature events --which future work would need to look into more quantitatively.

Reviewer #3 (Remarks to the Author)

The authors have sufficiently addressed the issues I had with the previous submissions. I recommend that this paper be accepted.

Response to Reviewers – Final Round

We want to thank all three reviewers for commenting on our manuscript in three versions and providing constructive comments that helped to substantially improve the manuscript. We provide responses to the remaining suggestions by reviewer #1 below.

Reviewer #1 (Remarks to the Author)

I believe the manuscript can now be published.

I commend the authors for their additional revisions, which meet my earlier request for adding clarifications with respect to the specific dynamics of high temperature impacts on cereals, distinguishing between average processes and local events linked to extreme conditions.

I would like to see two more minor additions to the final manuscript:

1. Some text that clarifies that the average crop dynamics of high temperature as mediated by phenology and water are well understood and have so for decades--albeit this paper undoubtedly adds significant numerical detail to tease which specific processes may have been determinant to yield over the historical period observed; and

Response:

We agree with the reviewer that such a statement helps to sharpen our contribution to the literature. We have added the following sentences to the discussion, based on the reviewer's suggestions:

"The applied statistical approach allows extracting average yield responses to exposure to different temperature bins across a large spatial area with varying small-scale management conditions. As such it is particularly suitable for the evaluation of GGCMs rather designed to reproduce yields responses on large scale than to resolve fine-scale variations in management. It adds to well-established knowledge of yield responses to temperature which is derived from field and chamber experiments. The application of GGCMs may help us to explore adaptation options on large scales."

2. Some text that mentions farmer adaptation under climate change. Again, it has been very well recognized and for decades that higher temperatures connected to climate change would have negative impacts on yields--sometimes even considerably. However, it has also been amply recognized that reporting negative yields by X% needs additional qualifiers...meaning they do not consider adaptation. In reality, before large impacts could set in, farmers would have shifted to cultivars better adapted to the new temperature regimes --within possibilities.

So some statement mentioning that the reported climate change results are valid without adaptation --would be more precise. Of course, the authors may further speculate on the fact that, adapted or

not--many cultivars would nonetheless suffer in future decades from an increased frequency of extreme temperature events --which future work would need to look into more quantitatively.

Response:

Following the reviewer's suggestion we have added the following statement to the discussion:

“Estimated yield responses under high levels of global warming should not be interpreted as predictions, since the GCM simulations do not commonly account for potential adaptation options. The implementation of management and thus adaptation options differs between models. For example, fertilizer application rates were held constant (PEGASUS, pDSSAT, pAPSIM) or adjusted flexibly according to nitrogen stress (EPIC-IIASA, EPIC-BOKU, GEPIC). The choice of cultivars was only allowed to change through time in PEGASUS, LPJ-GUESS and limitedly in GEPIC. Thus, the ensemble response to temperature exposure represents the average response across a range of different management assumptions. Individual farmer's options to adapt to more frequent temperature stress could dampen negative yield responses – though the extent may be limited (Lobell 2014, Schlenker 2009).”